# Periodic splay Fréedericksz transitions in a ferroelectric nematic

Bijaya Basnet [1,2], Sathyanarayana Paladugu [1], Oleksandr Kurochkin [3,4], Oleksandr Buluy[3], Natalie Aryasova [3], Vassili G. Nazarenko [3,4] ✉, Sergij V. Shiyanovskii [1,2] & Oleg D. Lavrentovich [1,2,5] ✉

Electric field-induced splay of molecular orientation, called the Fréedericksz transition, is a fundamental electro-optic phenomenon in nonpolar nematic liquid crystals. In a ferroelectric nematic $N_F$ with a spontaneous electric polarization **P**, the splay is suppressed since it produces bound electric charges. Here, we demonstrate that an alternating current (ac) electric field causes three patterns of $N_F$ polarization. At low voltages, **P** oscillates around the field-free orientation with no stationary deformations. As the voltage increases, the polarization acquires stationary distortions, first splay and twist in a stripe pattern and then splay and bend in a square lattice of +1 and -1 defects. In all patterns, **P** oscillates around the stationary orientations. The stationary bound charge is reduced by a geometrical "splay cancellation" mechanism that does not require free ions: the charge created by splay in one plane is reduced by splay of an opposite sign in the orthogonal plane.

Recent synthesis and characterization of liquid crystals[1–6] with large molecular dipoles established the existence of the ferroelectric nematic ($N_F$) with a spontaneous electric polarization **P** parallel to the long axes of the molecules and to the director $\hat{\mathbf{n}} \equiv -\hat{\mathbf{n}}$, which specifies the average quadrupolar molecular orientation[7]. The polarization **P** of the $N_F$ fluids shows a remarkable flexibility and sensitivity to external electric field, being free of the restrictions that the crystallographic axes impose on solid ferroelectrics. A very weak, ~$10^2$ V/m, field causes a twist of **P**, which holds a major promise for future technologies[7]. A similar realignment of the director $\hat{\mathbf{n}} \equiv -\hat{\mathbf{n}}$ of molecular orientation of a conventional apolar nematic (N), called the twist Fréedericksz transition, which enabled the modern industry of informational displays, requires fields 1000 times stronger[8]. In a striking contrast, the splay of molecular orientation is much more difficult to induce in the $N_F$ as compared to the N[5,7]. In a planar N cell, with molecules aligned parallel to the bounding plates, splay of $\hat{\mathbf{n}}$ occurs when an electric field **E** applied across the cell exceeds some threshold[8]. At the threshold, $\hat{\mathbf{n}}$ in the midplane of the cell starts to realign towards the vertical **E**-direction, provided the dielectric anisotropy of the N is positive, $\Delta\varepsilon = \varepsilon_\parallel - \varepsilon_\perp > 0$; here the subscripts refer to the mutual orientation of **E**

and $\hat{\mathbf{n}}$. Since the surface alignment keeps the molecules parallel to the plates, the ensuing deformation is splay; as the field increases, it is followed by splay-bend. When the same planar cell is cooled into the $N_F$ phase, the splay-bend Fréedericksz transition is strongly suppressed[7]. Chen et al. [7] made a direct comparison subjecting a planar cell of a thickness $d = 4.5\,\mu$m to a temperature gradient so that the N and $N_F$ regions coexisted. While the molecules in the N part started to tilt towards the field once the applied voltage of frequency 1 kHz reached a threshold of 1 V, the $N_F$ part showed no change in the optical appearance even when the field was increased to 3 V. This suppression of the splay-bend Fréedericksz effect is explained by block realignment of the polarization: a tilt of **P** deposits surface charges at the electrodes that screen the applied field[7,9].

In this work, by applying a battery of advanced optical techniques, we demonstrate that the splay Fréedericksz transition in the $N_F$ can be triggered by a high-frequency alternating current (ac) electric field. The scenario of this $N_F$ splay Fréedericksz transition turns out to be much richer than its non-polar N counterpart. An increasing voltage causes a succession of three polarization patterns. (i) At low voltages, without any threshold, **P** starts to oscillate up and down around the

[1]Advanced Materials and Liquid Crystal Institute, Kent State University, Kent, OH, USA. [2]Materials Science Graduate Program, Kent State University, Kent, OH, USA. [3]Institute of Physics, National Academy of Sciences of Ukraine, Kyiv, Ukraine. [4]Institute of Physical Chemistry, Polish Academy of Sciences, Warsaw, Poland. [5]Department of Physics, Kent State University, Kent, OH, USA. ✉e-mail: vnazaren@iop.kiev.ua; olavrent@kent.edu

initial planar orientation, preserving homogeneity within the cell and across the cell. (ii) An intermediate field on the order of $10^6$ V/m causes a periodic stationary splay and twists of **P** in the form of stripe patterns. (iii) At the highest voltage, **P** experiences stationary splay and bend in a periodic square lattice of +1 and −1 defects. In the splay-twist and splay-bend patterns, **P** oscillates around the stationary deformed directions. The frequency of oscillations in all states is the same as the frequency of the applied voltage, which means that they are caused by the polar interactions of **P** with the ac electric field. We attribute the cascade of patterns to the field-induced splay of **P** which produces bound charges. These bound charges can be screened by an electrostatic splay cancellation mechanism, in which the bound charge caused by the field-imposed vertical splay is reduced by a splay in the horizontal plane of the cell.

## Results

We explore the $N_F$ material RM734[1]. On cooling from the isotropic (I) phase, the phase sequence is I-188 °C-N-133 °C-$N_F$-84 °C-Crystal. Two glass plates with transparent indium-tin-oxide (ITO) electrodes and unidirectionally rubbed polyimide PI2555 layers form a planar cell. The rubbing direction $\mathbf{R} = (-1, 0, 0)$ along the negative direction of the $x$-axis in the $xy$-plane of the sample aligns $\mathbf{P} = (-P, 0, 0)$ in the $N_F$ and $\hat{n}$ in the N. The ac field is applied along the $z$-axis, $\mathbf{E} = (0, 0, \pm E)$. The frequency is $f = 200$ kHz, square or sinusoidal wave.

### Homogeneous splay Fréedericksz transition in the N phase
The N phase shows a typical splay Fréedericksz transition with a well-pronounced threshold $U_{N,th}$, Fig. 1a. The threshold is low, $U_{N,th} = 0.24$ V, Fig. 1b, c. When the voltage exceeds $U_{N,th}$, $\hat{n}$ tilts away from the original (voltage-free) direction along the $x$-axis. As predicted by the theory[10], the tilt angle does not depend on in-plane coordinates $x$ and $y$ but varies along the $z$-axis normal to the cell and grows with the voltage: $\psi(z, U) = \psi_m(U) \sin\frac{\pi z}{d}$, where $0 \le z \le d$, $d$ is the cell thickness, $\psi_m(d/2, U) = A\sqrt{\frac{U}{U_{N,th}} - 1}$ is the maximum tilt in the midplane, and $A = \frac{2}{\sqrt{\frac{K_{33}}{K_{11}} + \frac{\varepsilon_\parallel}{\varepsilon_\perp} - 1}}$ is the numerical factor defined by the ratio of bend-to-splay elastic constants $\frac{K_{33}}{K_{11}}$ and by dielectric permittivities. We find $A$ by fitting the voltage dependence of retardance $\Gamma(U) = \int_0^d \left( \frac{n_e n_o}{\sqrt{n_e^2 \sin^2 \psi(z, U) + n_o^2 \cos^2 \psi(z, U)}} - n_o \right) dz$ immediately above $U_{N,th}$, using the independently determined $d$ as well as the ordinary $n_o$ and extraordinary $n_e$ refractive indices, Supplementary Fig. 1. The fitting, Fig. 1c, yields $A = 0.8$ and $\frac{K_{33}}{K_{11}} + \frac{\varepsilon_\parallel}{\varepsilon_\perp} = 7.3$. Since $\frac{\varepsilon_\parallel}{\varepsilon_\perp} > 1$, one expects $\frac{K_{33}}{K_{11}} \approx 5 - 6$, which agrees with the data reported by Mertelj et al. [3] for the N phase of RM734. With the reported[3] $K_{11} = 3.0$ pN, the threshold voltage $U_{N,th} = \pi \sqrt{\frac{K_{11}}{\Delta \varepsilon \varepsilon_0}} = 0.24$ V yields $\Delta \varepsilon = 58$ at $f = 200$ kHz, a value

that is about 5 times larger than the ones reported for the conventional N materials such as pentylcyanobiphenyl[8].

The realignment of the N director is caused by the dielectric anisotropy coupling with the free energy density $\left[ -\frac{1}{2} \Delta \varepsilon \varepsilon_0 (\mathbf{E} \cdot \hat{n})^2 \right]$ and, thus, is not sensitive to the polarity of the field. The response of the $N_F$ phase to the electric field is very different, since it involves also a linear term $(-\mathbf{E} \cdot \mathbf{P})$ in the free energy density. As described below, the polarization **P** of the $N_F$ phase shows two modes of response to an ac electric field: (1) fast oscillations with the frequency of the field and (2) stationary deformations of the splay-twist and splay-bend type, at which these oscillations are centered. Accordingly, we introduce two types of notations. The instantaneous tilt angle of **P** with respect to the $x$-axis is denoted $\psi$, while its time-averaged (stationary) value is denoted $\bar{\psi}$; similar notations with the bar are preserved for other variables, such as the optical retardance, $\Gamma$ and $\bar{\Gamma}$. We first describe the response that is homogeneous in the $xy$ plane of the cell, Fig. 2, in which case the polarization oscillates without any threshold around its original (field-free) planar orientation.

### Homogeneous oscillatory response of the planar $N_F$ cell
**(A) Frequency and amplitude of oscillations.** The electric scheme used to analyze the electro-optic response of $N_F$ cells is shown in Fig. 2a. The oscillations of **P** reveal themselves in the oscillations of the transmitted light intensity through the cell and two crossed polarizers (oriented at 45° to the rubbing direction **R**) since **P** is collinear with the $N_F$ optic axis, Fig. 2b, c. To distinguish when the polarization tilts up and down, the cell is tilted by 15° around the $y$-axis to yield an oblique incidence of the probing laser beam, Supplementary Fig. 2. Once the ac sinusoidal voltage $U = \sqrt{2} U_{rms} \sin(2\pi f t)$ of the root mean square amplitude $U_{rms}$, frequency $f = 200$ kHz and period $T = 5 \mu s$ is applied, the normalized and relative (with respect to the field-free state) transmitted light intensity $\tau[U(t)]$ oscillates with the same frequency as the frequency of the applied voltage, Fig. 2b, c. The function $\tau[U(t)]$ is expressed through the field-dependent optical retardance $\Gamma(U)$, where $U = U(t)$ is the function of time $t$ and the retardance $\Gamma_0$ in the absence of the field,

$$\tau(U) = \sin^2\left\{ \frac{\pi}{\lambda} \left[ \Gamma(U) + \Gamma_0 \right] \right\} - \sin^2\left( \frac{\pi}{\lambda} \Gamma_0 \right)$$
$$= \frac{1}{2} \sin\left[ \frac{2\pi}{\lambda} \Gamma(U) \right] \sin\left( \frac{2\pi}{\lambda} \Gamma_0 \right) + \sin^2\left[ \frac{\pi}{\lambda} \Gamma(U) \right] \cos\left( \frac{2\pi}{\lambda} \Gamma_0 \right).$$

Both $\Gamma(U)$ and $\Gamma_0$ represent a sum of the retardances of a cell and an optical compensator. In the experiment, to increase the sensitivity to the field-induced changes, we select optical compensators in such a way that $\frac{2\pi}{\lambda} \Gamma_0 = \frac{\pi}{2}$, so that the second term in the last formula disappears and $\tau[U(t)] = \frac{1}{2} \sin\left\{ \frac{2\pi}{\lambda} \Gamma[U(t)] \right\}$. Since the changes in the oblique incidence are small, a few percent, Fig. 2c, the equation simplifies to

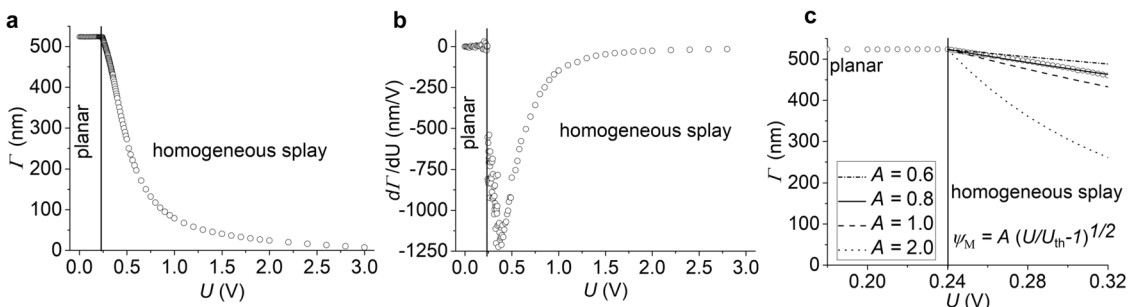

**Fig. 1 | Homogeneous splay Fréedericksz transition in the N phase of RM734. a** Optical retardance vs applied voltage; $d = (2.9 \pm 0.1)$ μm; $f = 200$ kHz, square wave; 170 °C. **b** Rate of retardance change demonstrating a threshold at 0.24 V. **c** Fitting the voltage dependence of retardance $\Gamma(U)$ yields $A = 0.8$.

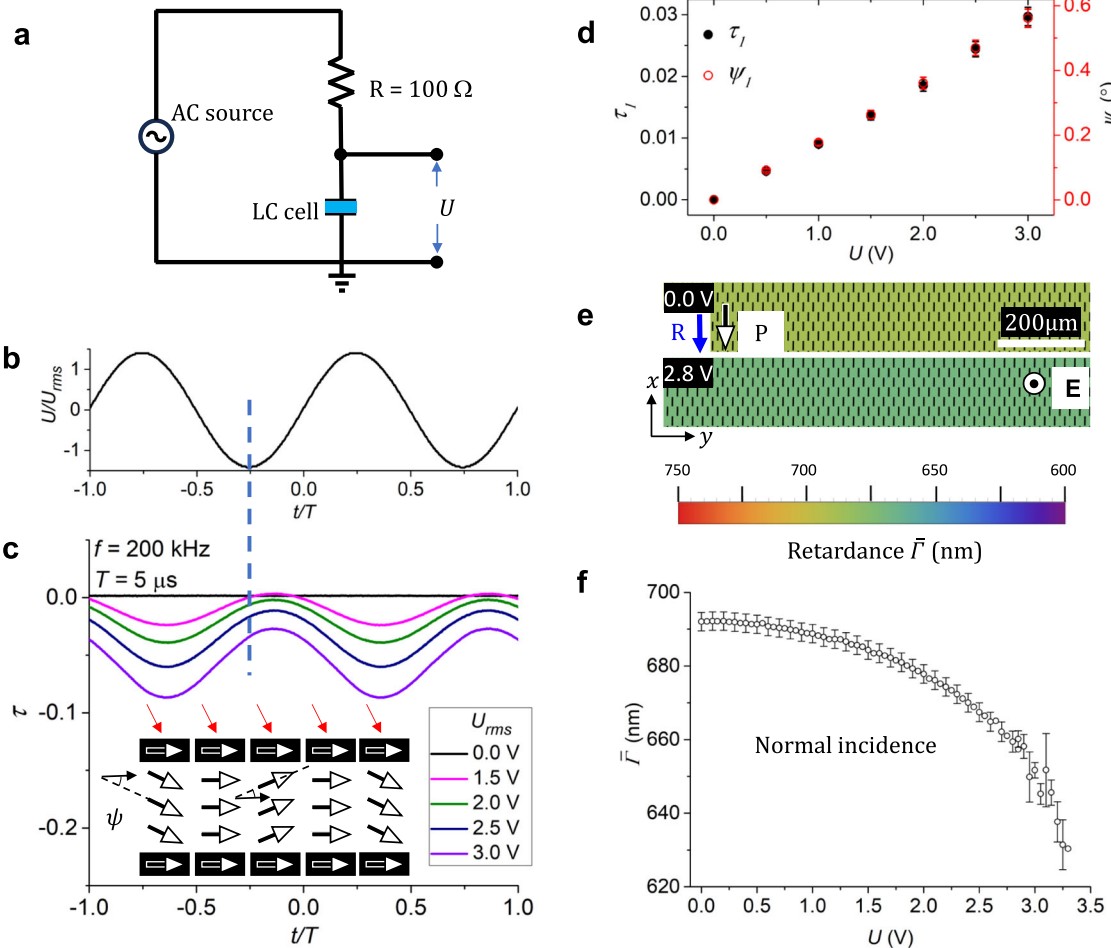

**Fig. 2 | Homogeneous oscillations of polarization in planar $N_F$ cell at low voltages. a** Electric circuit to measure the electro-optical response of the $N_F$ cells. **b** Applied sinusoidal voltage profile; 200 kHz, period $T = 5\,\mu s$. **c** Transmitted light intensity vs time for obliquely incident light beam of a wavelength 633 nm. Up and down realignments of the polarization **P** (arrows with white heads) by the same angle cause a different optical retardance for obliquely incident light beam. $d = (3.0 \pm 0.1)\,\mu m$, 125 °C. **d** Amplitudes of the first harmonic of transmitted intensity $\tau_1$ and of polarization tilt $\psi_1$ are proportional to each other and to the applied voltage; error bars correspond to the accuracy of refractive indices measurement and standard deviation of the cell thickness. **e** PolScope Microimager maps of homogeneous time-averaged optical retardance $\bar{\Gamma}$ for normal incidence of light of a wavelength 535 nm; the black ticks show the optical axis calculated by the PolScope. **R** shows the direction of rubbing at both plates. 200 kHz, square waveform. $N_F$ cell with $d = (2.7 \pm 0.1)\,\mu m$; 125 °C. **f** Time-averaged retardance $\bar{\Gamma}$ vs voltage; the same cell; error bars are standard deviations.

$\tau = \frac{\pi}{\lambda}\Gamma[U(t)]$. Introducing the notation $\delta = \frac{n_e^2 - n_o^2}{2n_o^2} \approx \frac{n_e - n_o}{n_o}$, one can write the retardance $\Gamma(\psi)$ of the $N_F$ cell as a function of the tilt $\psi$ of **P** away from the $x$-axis. The tilt is a sum of its stationary value $\bar{\psi}$ and an oscillating term of an amplitude $\psi_1$: $\psi = \bar{\psi} + \psi_1 \sin(2\pi f t - \phi_1)$; here $\phi_1$ is the phase shift clearly seen in Fig. 2b, c. The stationary tilt $\bar{\psi}$ should be close to zero in a planar cell; an independent experiment demonstrates that the in-plane electric field does not change the retardation. The dependence $\Gamma(\psi)$ reads (see "Methods")

$$\Gamma(\psi) = \frac{\delta n_o d}{(1 + 2\delta \sin^2\psi)} \left\{ \sin\beta \sin 2\psi + 2 \frac{(1 + 2\delta)\sin^2\psi + \cos^2\beta \left[1 - 2\sin^2\psi\left(1 + 2\delta \sin^2\psi\right)\right]}{\cos\beta (1 + 2\delta \sin^2\psi) + \sqrt{(1 + 2\delta)\left(\cos^2\beta + 2\delta \sin^2\psi\right)}} \right\} \quad (1)$$

where $\beta$ is the incidence angle, measured with respect to the $z$-axis in the $xz$ plane. The retardance is comprised of two terms, which are, respectively, odd and even functions of $\beta$ and $\psi$. The amplitudes of first harmonics of light transmittance $\tau_1 = (2\pi\delta n_o d\psi_1/\lambda)\sin\beta$ and of retardance $\Gamma_1 = 2\delta n_o d\psi_1 \sin\beta$ are defined by the first term in Eq. (1). These amplitudes allow one to find the amplitude $\psi_1$ of the polarization tilt since the other parameters are known: $\beta = 15°$, $n_o = 1.49$, $n_e = 1.73$ (both measured at 633 nm, Supplementary Fig. 1), and

thickness $d = 3.0\,\mu m$. We find that both $\tau_1$ and $\psi_1$ are small and proportional to the applied voltage, Fig. 2d. In particular, $\psi_1 = 0.2°$ at $U = 1\,V$, Fig. 2d. A similar value of $\psi_1$ is obtained by balancing the electric and viscous torques acting on the polarization: $\psi_1 = \frac{J_1}{2\pi f P}$, where $J_1 = I_1/\Sigma$ is the current density between the electrodes of an area $\Sigma$; $I_1$ is the amplitude of the total ac current in the circuit. With the measured $\Sigma = 1\,cm^2$ and $J_1 = 3.8 \times 10^2\,A/m^2$ at $U = 1\,V$, one estimates $\psi_1 \approx 0.005$ rad $\approx 0.3°$, which is close to the experimental results in Fig. 2d.

For small amplitude $\psi_1$ of oscillations, the expected decrease of retardance under normal incidence, $\beta = 0$, is only about $1 - \Gamma/\Gamma_0 \approx \psi_1^2 \approx 2.5 \times 10^{-5}$. The experimentally measured change of retardance, Fig. 2e, f, and Supplementary Fig. 3, is much larger, $1 - \Gamma/\Gamma_0 \approx 0.04$. The discrepancy is explained by the electric heating of the cell. The temperature of the $N_F$ cell increases by as much as 3 °C when subject for ~10 s to an applied ac voltage 2 V of 200 kHz frequency; the heating time is shorter than the typical duration of the electro-optical experiment. As the voltage increases, so does the temperature: it reaches 7 °C degrees for 4 V and 18 °C for 6 V; the temperature increase is quantified by observing the phase transition points of the cells under the applied voltage. A temperature change from 125 °C to 130 °C causes a change of birefringence from $n_e - n_o = 0.235$ to

$n_e - n_o = 0.218$; the ratio $(1 - 0.218/0.235)(3\,°C/5\,°C) \approx 0.04$, is the same as $1 - \Gamma/\Gamma_0 \approx 0.04$ in Fig. 2e, f, thus confirming that the main contribution to the retardance change is the electric heating and not the oscillations of the optic axis.

**(B) Uniformity of polarization tilt.** Below we demonstrate that the tilt angle $\psi$ of homogeneous oscillations does not depend on the $z$-coordinate normal to the cell. We introduce the complex amplitudes' representation of the oscillating angle $\psi = \bar{\psi} + \Psi_1 \exp(-i\omega t) + \Psi_{-1} \exp(i\omega t)$, where $\omega = 2\pi f$ is the angular frequency and $\Psi_{-1}$ is the complex conjugate of $\Psi_1 = \frac{i}{2}\psi_1 \exp(i\phi_1)$; similar representations are also used below for the electric field $\mathbf{E} = (0, 0, E)$, electric displacement $\mathbf{D}$, and the current density $\mathbf{J}$. With this notation, the oscillations of the tilt angle are controlled by the balance of the electric, viscous, and elastic torques, written as

$$PE_1 \cos\bar{\psi} + i\omega\gamma\Psi_1 + K_{eff}(\bar{\psi})\Psi_1'' = 0, \qquad (2)$$

where $E_1$ is the complex amplitude of the first harmonic of the field, $\gamma$ is the rotational viscosity (usually denoted $\gamma_1$ but we omit the subscript to avoid confusions with the harmonics counting), $K_{eff}(\bar{\psi}) = K_{11} + (K_{33} - K_{11})\sin^2\bar{\psi}$ is the combination of the splay $K_{11}$ and bend $K_{33}$ elastic constants, double prime is the second derivative with respect to $z$. An additional relation between $E_1$ and $\Psi_1$ follows from the first Maxwell equation:

$$\text{div}\left(\dot{\mathbf{D}}(t) + \mathbf{j}(t)\right) = \text{div}\mathbf{J}(t) = 0, \qquad (3)$$

where the dot means the time derivative, $\mathbf{D}(t) = \mathbf{P}(t) + \varepsilon_0\varepsilon_\perp\mathbf{E}(t) + \varepsilon_0\Delta\varepsilon\,\hat{\mathbf{n}}(t)(\hat{\mathbf{n}}\cdot\mathbf{E}(t))$ is the electric displacement written in the assumption that there is no dispersion; $\mathbf{j} = \sigma_\perp\mathbf{E} + (\sigma_\parallel - \sigma_\perp)\hat{\mathbf{n}}(\hat{\mathbf{n}}\cdot\mathbf{E})$ is the ohmic current density caused by the free ionic charges, $\sigma_\parallel$ and $\sigma_\perp$ are the conductivities of the $N_F$ along and perpendicularly to the director, respectively. $\mathbf{J} = \dot{\mathbf{D}} + \mathbf{j} = \bar{\mathbf{J}} + \mathbf{J}_1 \exp(-i\omega t) + \mathbf{J}_{-1} \exp(i\omega t)$ is the total current density that includes also the shifts of bound charges. In the homogeneous structure of Fig. 2, the first harmonic of the current density $\mathbf{J}$ has no components in the $xy$ plane, and its z-component $J_1$ must be constant, $J_1 = $ const, since div $\mathbf{J} = 0$. The latter condition provides the second equation for $\Psi_1$ and $E_1$:

$$J_1 = -i\omega\left(P\Psi_1\cos\bar{\psi} + \varepsilon_0\widetilde{\varepsilon}_{eff}(\bar{\psi})E_1\right), \qquad (4)$$

where $\widetilde{\varepsilon}_{eff}(\bar{\psi}) = \widetilde{\varepsilon}_\perp(\omega) + [\widetilde{\varepsilon}_\parallel(\omega) - \widetilde{\varepsilon}_\perp(\omega)]\sin^2\bar{\psi}$, and $\widetilde{\varepsilon}_\mu(\omega) = \varepsilon_\mu(\omega) + i\sigma_\mu/\omega\varepsilon_0$, $\mu = \parallel, \perp$. Using Eq.(4), we eliminate $E_1$ from Eq. (2) and obtain the equation for $\Psi_1$ which has the solution that satisfies the boundary condition $\Psi_1(z = 0, d) = 0$ and writes

$$\Psi_1(z) = \Psi_1^{bulk}\left(1 - \frac{\cosh(z - d/2)/\xi}{\cosh\frac{d}{2\xi}}\right), \qquad (5)$$

where

$$\psi_1^{bulk} = \frac{iPJ_1\cos\bar{\psi}}{\omega\left(P^2\cos^2\bar{\psi} + i\omega\gamma\varepsilon_0\widetilde{\varepsilon}_{eff}(\bar{\psi})\right)} \qquad (6)$$

is the tilt angle in the bulk and

$$\xi = \sqrt{K_{eff}(\bar{\psi})\varepsilon_0\widetilde{\varepsilon}_{eff}(\bar{\psi})/\left(P^2\cos^2\bar{\psi} + i\omega\gamma\varepsilon_0\widetilde{\varepsilon}_{eff}(\bar{\psi})\right)} \qquad (7)$$

is the characteristic length which defines the range of splay deformations of $\mathbf{P}$ near each of the two substrates. This thickness can be estimated as[9] $\xi = \sqrt{\frac{\varepsilon_0\varepsilon K_{11}}{P^2}}$, where $\varepsilon \sim 10 - 100$, refs. [11,12],

$P = 6 \times 10^{-2}$ C m$^{-2}$, ref. [7], while $K_{11}$ might be in the range 10–100 pN, ref. [13]. With these estimates, one finds $\xi = (0.5 - 5)$ nm, of a tiny molecular scale. The strongly splayed $\xi$-regions could exist only when the zenithal surface anchoring coefficient is large, $W > K_{eff}(\bar{\psi})/\xi = 2 \times 10^{-2}$ J m$^{-2}$. A weaker anchoring does not restrict the values $\Psi_1(z = 0, d)$ and $\Psi_1(z) = \Psi_1^{bulk}$ along the entire z-axis, without any inhomogeneities near the substrates. With a high accuracy, $\Psi_1(z) = \Psi_1^{bulk}$, i.e., is z-independent, and the elastic torque in Eq. (2) can be neglected. The model proposed by Clark et al.[9] for "block reorientation" of $\mathbf{P}$ by a dc field exhibit a similar z-independence of the tilt and its linear growth with the voltage, as in Fig. 2d.

**(C) Phase of oscillations.** The consideration above also explains the phase shift between the voltage and the optical response in Fig. 2b, c. Equation (6) shows the phase shift between $J_1$ and $\Psi_1$. The phase shift between $J_1$ and the voltage $U_1$ applied to the cell can be obtained from the Kirchhoff equation

$$U_1 = I_1\left(Z_1^{(NF)} + 2Z_1^{(sl)} + R^{(el)}\right), \qquad (8)$$

where

$$Z_1^{(NF)} = \frac{E_1 d}{J_1\Sigma} = \frac{\gamma d}{\left(P^2\cos^2\bar{\psi} + i\omega\gamma\varepsilon_0\widetilde{\varepsilon}_{eff}(\bar{\psi})\right)\Sigma} \qquad (9)$$

is the complex impedance at the angular frequency $\omega$ of the $N_F$ layer; $Z_1^{(sl)} = i\, d_{sl}/(\omega\varepsilon_0\varepsilon_{sl}\Sigma)$ is the corresponding impedance of the surface alignment layer (such as a polymer) with the dielectric constant $\varepsilon_{sl}$ and thickness $d_{sl}$, respectively; $R^{(el)} = \eta G$ is the resistance of cell electrodes of the area $\Sigma$, $\eta$ is the electrode's surface resistivity and $G$ is the dimensionless factor defined by the electrodes' shape. Equations (6) and (8) explain that $I_1$ mediates the phase shift between $U_1$ and $\Psi_1$ observed in Fig. 2b, c.

In the homogeneous oscillatory state described so far, $\mathbf{P}$ realigns towards $\mathbf{E}$ always remaining in the vertical $xz$ plane formed by the rubbing direction $\mathbf{R}$ and $\mathbf{E}$, Fig. 2e. There are no hydrodynamic flows in the system, as verified by doping the material with fluorescent spheres of a diameter 300 nm. As the voltage increases above some threshold $U_{ST}$, which is about 2.8 V for the $d = 2.7$ µm cell, the polarization $\mathbf{P}$ starts to deviate from the $\mathbf{RE}$ plane, forming stripe domains, Fig. 3 and Supplementary Movies 1,2, as described below.

## Periodic splay-twist polarization pattern of stripes

Above $U_{ST}$, the $N_F$ develops periodic modulation of $\mathbf{P}$ along the $y$-axis perpendicular to $\mathbf{R}$, Fig. 3a, which means a non-vanishing derivative of the $y$-component of polarization, $\frac{\partial P_y}{\partial y} \neq 0$. This derivative contributes to the bound charge $\rho_b = -\left(\frac{\partial P_y}{\partial y} + \frac{\partial P_z}{\partial z}\right)$. The bound charge can be reduced if the polarization also develops a $z$-dependence $\frac{\partial P_z}{\partial z} \neq 0$, provided $\frac{\partial P_y}{\partial y}\frac{\partial P_z}{\partial z} < 0$. Polarizing microscopy demonstrates that the modulations are comprised of predominantly splay (S) regions, Fig. 3a, b, intercalated between twist regions with left- and right-handedness (TL and TR, respectively), Fig. 3c, d. Observations with obliquely propagating light, Fig. 3e, f, prove that in an S region, a positive splay $\frac{\partial P_y}{\partial y}$ in the horizontal $xy$ plane coexists with the negative splay $\frac{\partial P_z}{\partial z}$ in the vertical $xz$ plane (and a negative $\frac{\partial P_y}{\partial y}$ is accompanied with a positive $\frac{\partial P_z}{\partial z}$) so that the condition of splay cancellation $\frac{\partial P_y}{\partial y}\frac{\partial P_z}{\partial z} < 0$ is always fulfilled in all the S-regions.

In the center of each S region, the optical axis is in the $\mathbf{RE}$ plane, Fig. 3a, b. Around this center line, $\mathbf{P}$ develops an in-plane splay that resembles a letter "V" or a letter "Λ" when viewed along the $x$-axis. We

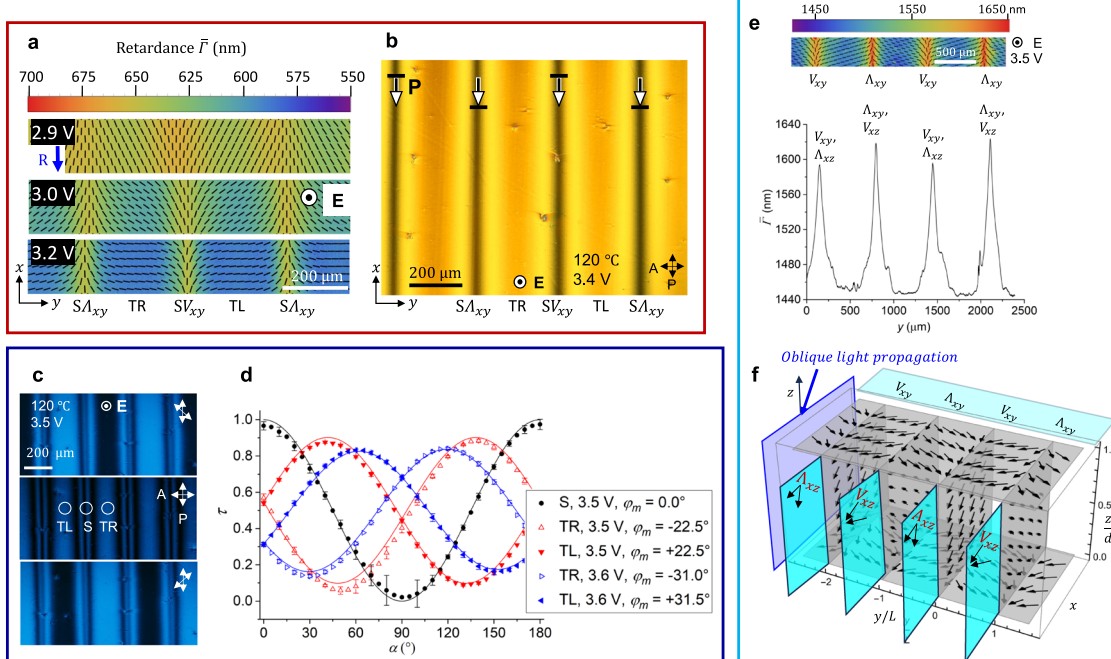

**Fig. 3 | Periodic splay-twist stripe domains in planar N$_F$ cell at moderate voltages. a** PolScope Microimager texture of stripes with splay regions S; the splay of polarization **P** in the $xy$ plane resembles alternating letters "V" and "Λ". **b** Polarizing microscopy shows extinct splay S-regions and bright twist regions of left-handed twist (TL) and right-handed twist (TR). Extinct centers of the S regions prove that **P** is in the **RE** plane there. The head of a nail attached to each tilted **P** vector in the S regions indicates that this end of **P** is closer to the observer. **c** Observations with uncrossed polarizers show complementary contrasts of TL and TR regions; monochromatic 485 nm light. **d** Transmitted light intensity vs. the angle $\alpha$ between the polarizer and analyzer shows no twist in the S regions and left- and right-handed twists in TL and TR, respectively. Solid lines are numerical fits of the data. The maximum twist angle $\bar{\varphi}_m$ between **P** and the $x$-axis grows with the voltage. Error bars in (**d**) show the amplitude of light intensity fluctuations. N$_F$ cell in (**a**–**d**) with $d = (3.1 \pm 0.1)\,\mu m$; 120 °C; 200 kHz, square waveform. **e** Top of the cell is tilted towards the observer by 15°; PolScope Microimager shows optical retardance higher in the splay $\Lambda_{xy}$ regions than in the $V_{xy}$ regions. **f** Splay-canceling geometry of the S-regions explains the difference in the retardance in (**e**): the horizontal $\Lambda_{xy}$ splay is accompanied by the vertical splay $V_{xz}$ which appears more planar and thus of a higher retardance for the obliquely propagating light; in contrast, $\Lambda_{xz}$ appears more "homeotropic" and of a weaker retardance. Each horizontal splay is accompanied by the vertical splay of an opposite sign so that the splay cancellation condition $\frac{\partial P_y}{\partial y}\frac{\partial P_z}{\partial z} < 0$ is fulfilled. In (**e**, **f**), $d = (6.5 \pm 0.1)\,\mu m$; 125 °C. 200 kHz, sinusoidal waveform, 475 nm light.

label these splays as S$V_{xy}$ and S$\Lambda_{xy}$, respectively, in Fig. 3a, b, and $V_{xy}$ and $\Lambda_{xy}$ in Fig. 3e, f. When the cell is tilted around the $y$-axis, the neighboring S regions show different interference colors, Supplementary Fig. 4, which indicates that **P** acquires a stationary time-independent tilt $\bar{\psi} \neq 0$ along the $z$-axis, which alternates in sign from one S-region to the next, Fig. 3b, e, f.

The regions TL and TR, separated by an S region, exhibit a twist of **P** around the normal to the cell, as evidenced by the dependence of transmitted light intensity $\tau(\alpha)$ on the angle $\alpha$ between the polarizer P and the analyzer A, Fig. 3c, d, and by fluorescent confocal polarizing microscopy[14], Supplementary Fig. 5. The dependence $\tau(\alpha)$ reaches a minimum at $\alpha \neq 90°$, while in the S-region, the minimum is at $\alpha = 90°$, Fig. 3d. These minima in TL and TR are not as deep as in the S-regions. The linear polarization of light at the entry becomes elliptical within the cell because birefringence is high, ~0.25, and the optic axis twists. An elliptically polarized light beam cannot be entirely extinct by the analyzer. Symmetry of the curves $\tau(\alpha)$ indicates that **P** is left- and right-handed twisted in the TL and TR regions, respectively, Fig. 3b, c. Numerical fitting of $\tau(\alpha)$ performed as described in ref. 15, assuming that **P** twists uniformly along the $z$-axis, making a stationary angle $\bar{\varphi} = \bar{\varphi}_m \cos\left(\frac{\pi z}{d}\right)$ with **R**, shows that the amplitude $\bar{\varphi}_m$ grows with the voltage, Fig. 3d, from $\bar{\varphi}_m \approx 22.5°$ at 3.5 V to $\bar{\varphi}_m \approx 32°$ at 3.6 V. In the centers of S-regions, there is no twist, $\bar{\varphi}_m = 0$.

The period $2L$ of stripes is about 100–300 times larger than the cell thickness $d$ and grows with $d$, temperature, and inverse frequency $1/f$ but depends little on $U$, Supplementary Fig. 6. The threshold $U_{ST}$ is a non-monotonous function of $f$ with a minimum ($\approx 2\,V$) at $f =$ 200 kHz, Supplementary Fig. 7, and decreases with the cell thickness $d$, Supplementary Fig. 8.

Time-resolved measurements of the cell's optical transmittance for an obliquely incident light beam making 15° with the normal to the cell, confirm that the polarization **P** at $U \geq U_{ST}$ continues to oscillate with the frequency of the applied electric field, Fig. 4a–c. The oscillations are centered at the stationary tilt $\bar{\psi}$, the sign of which alternates from one S-region to the next, as confirmed by dynamics of light transmittance in Fig. 4c. There are no hydrodynamic flows, Supplementary Movie 2, similarly to the case of homogeneous oscillations at lower voltages, Supplementary Movie 1. Note that there is some phase shift between the applied voltage and the optical response in Fig. 4b, similarly to the case presented in Fig. 2, for a qualitatively similar reason. However, the stripe pattern is strongly inhomogeneous, and the analysis of the shift is less transparent and will require additional studies.

In Fig. 4c, S1 and S3 regions show a nearly identical behavior; so do S2 and S4 regions. The overall transmittance of light alternates from one S region to the next. The transmittance of S1, S3 is lower than the transmittance of S2, S4 since the obliquely incident light probes oscillations centered at the stationary tilt $\bar{\psi} < 0$ in S1, S3, and $\bar{\psi} > 0$ in S2, S4, as illustrated in Fig. 4d. For the probing beam, the states S1, S3 appear to be more "homeotropic" and states S2, S4 more "planar", similarly to the effect illustrated in Fig. 3e, f for the S$V_{xz}$ and S$\Lambda_{xz}$ regions.

To estimate the stationary tilt $\langle \bar{\psi} \rangle = \frac{1}{2}\arcsin\left[\frac{1}{d}\int_0^d \sin(2\bar{\psi})dz\right]$ averaged over the $z$-axis, we apply Eq. (1) to the difference of optical retardances in points S1 and S2, Fig. 4a, c, in which the tilts are of an opposite sign: $\Gamma_{S1} - \Gamma_{S2} = 2\delta n_o d \sin\beta \sin 2\langle \bar{\psi} \rangle$. Equation (1) is

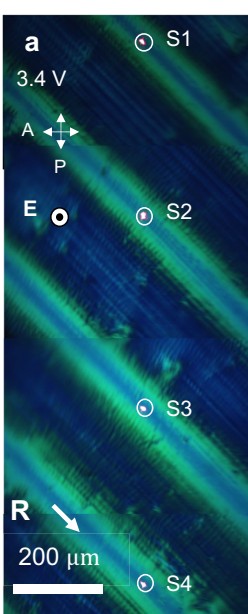
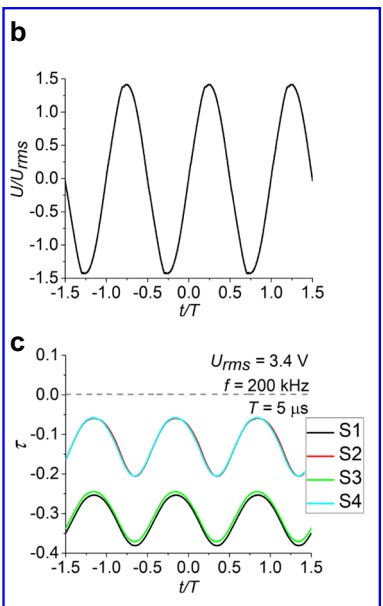
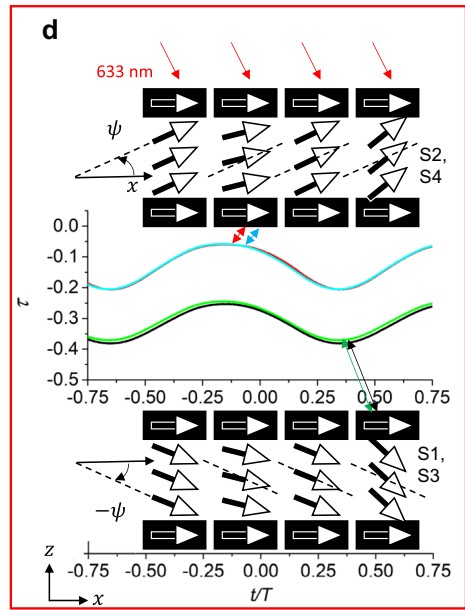

**Fig. 4 | Dynamics of the periodic splay-twist stripe domains in planar $N_F$ cell at moderate voltages. a** Polarizing microscopy of splay-twist domains with four S-regions; the focused laser beam tests the dynamic response of transmitted light intensity at four circled locations, S1–S4. **b** Applied voltage profile; 200 kHz, sinusoidal waveform, $U_{rms}$ = 3.4 V, period $T$ = 5 μs. **c** Transmitted light intensity vs time for a light beam propagating at 15° to the normal to the cell. Dashed line corresponds to zero applied voltage. **d** Schematic oscillations of **P** around stationary tilts that alternate in sign from one S-region to the next. $N_F$ cell with $d$ = (3.0 ± 0.1) μm, 125 °C.

applicable to the S-regions since these regions are not optically active. The effect of heating and the quadratic terms such as $\sin^2 \psi$ in Eq. (1) do not contribute. Using the values of $\delta$, $n_o$, and $d = 3$ μm to fit the data in Fig. 4c, one finds $\langle \bar{\psi} \rangle = 5.4°$.

**Periodic splay-bend pattern with a square lattice of defects**

As the voltage increases above another threshold $U_{SB}$, which ranges between 3.5 V and 8.5 V, depending on the frequency and cell thickness, Supplementary Figs. 7, 8, stripe domains suffer a complete reconstruction into a square lattice of +1 and −1 defects, Figs. 5, 6, Supplementary Figs. 9, 10, and Supplementary Movie 3. The reconstruction involves transient microstripes of a width ~10 μm that develop in the S-regions and are orthogonal to them. In the established square lattice, the polarization experiences splay and bend but no twist, Fig. 5. The retardance $\Gamma$ is 4–7 times lower than the retardance of the planar state and approaches zero at the cores of +1 and −1 defects, Fig. 5b, e, which implies that **P** is mostly along the $z$-axis. The in-plane polarization field around the +1 defects is radial, with a strong splay of **P**. The textures are thus dramatically different from +1 circular vortices and "closure loops" with bend of **P** in solid ferroelectrics[16,17] and from +1 bend vortices in $N_F$ films with degenerate anchoring[18–20].

The transmitted intensity shows oscillations with the frequency of the applied electric field, Fig. 6a–d, which develop only around the stationary directions of polarization that have an in-plane component. An interesting distinctive feature of the periodic square lattice of +1 and −1 defects is that the potential difference measured at the cell's electrodes acquires a strong static component and shows an unusual saw-tooth profile, Fig. 6c, despite the fact that the generator produces a standard sinusoidal waveform with a zero dc bias. In the homogeneous oscillating state, this static component is absent, and the voltage profile retains its sinusoidal shape, but in the square lattice state, the voltage measured across the cell is of a biased triangular shape, with its static component being of the same order of magnitude as the voltage amplitude, Fig. 6c. The static component indicates that the $N_F$ slab in the splay-bend state acquires uncompensated charges, positive at one plate and negative at the other plate. A related feature is that the high-frequency oscillations of polarization in the square

lattices are accompanied by hydrodynamic flows, visualized by fluorescent spherical tracers of the diameter 300 nm, Fig. 6e, f; Supplementary Movie 4. The in-plane velocities are low near the +1 cores and high, on the order of 100 μm/s near the −1 cores, Fig. 6f, which suggests that the 3D flows form loops with the z-component along the +1 cores. This hydrodynamics is likely to be caused by the uncompensated charges at the top and bottom surfaces of the $N_F$, which create a dc voltage ~(5–10) V, Fig. 6c; these voltages are typical for the onset of electrohydrodynamic flows in conventional nematics[21].

The large-period (as compared to the slab thickness) stripes and square pattern of splay-bend resemble the stripes and square lattices in field-free hybrid aligned N films[22] and in the geometry of the bend Fréedericksz transition in a material of a negative $\Delta \varepsilon$ intentionally doped with a large amount of ions[23]. In the latter case, the patterns are observed below some limiting frequency of the applied field, which is a few hundreds of Hz, a thousand times lower than in our case. The stripes appear at higher voltages than the square lattices; the behavior is opposite in our case. The periodicities are about one order of magnitude lower than in our case. We attribute the differences to the different nature of operational charges in the two works. Sasaki et al.[23] explain the observed patterns by the nonuniform distribution of freely moving ions that accumulate at the insulating polymer alignment layers. In our case, the main driving mechanism are bound charges which in the case of stripes tend to geometrically self-compensate and in the case of the square lattices, produce a stationary potential difference between the electrodes.

Further increase of the voltage above 9 V produces chaotic electrohydrodynamic flows and a transition to the homeotropic N state (caused by electric heating), that destroy the square lattice, Supplementary Movies 5, 6.

## Discussion

Figure 7 shows the snapshots of the three stages of the $N_F$ splay Fréedericksz transition suggested by the experiments: (a) oscillating homogeneous tilt of polarization, (b, c) stationary periodic splay-twist with oscillations of **P** around these stationary deformations, and (d, e) stationary periodic splay-bend with oscillations of **P** and

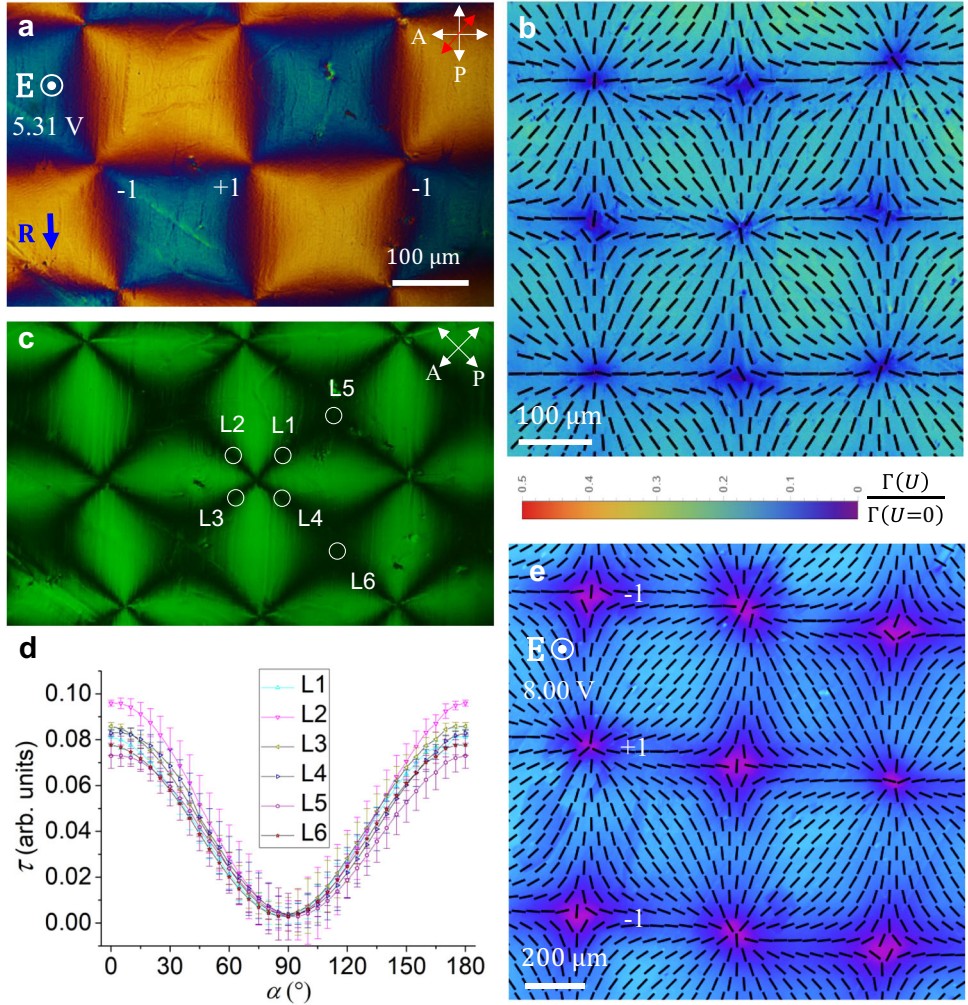

**Fig. 5 | Splay-bend periodic Fréedericksz transition in a planar $N_F$ cell.**
**a** Polarizing microscopy texture of $d = (2.2 \pm 0.1)\,\mu m$ cell, observed with a 550 nm optical compensator (the slow direction is along the red arrow) with a square array of +1 and −1 defects. **b** The same cell, PolScope Microimager mapping of the optical axis and retardance at 655 nm probing light. **c** The same, observations with rotated crossed polarizers in monochromatic 532 nm light. **d** Transmitted light intensity through six locations L1–L6 in (**c**) as a function of the angle $\alpha$ between the polarizer and analyzer; the minimum intensity at $\alpha = 90°$ proves the absence of twist. Error bars show the amplitude of light intensity fluctuations. **e** PolScope Microimager mapping of the optical axis (ticks) and retardance of a cell with $d = (6.6 \pm 0.1)\,\mu m$; 655 nm probing light. The pseudocolor bar shows how much smaller the retardance in (**b**, **e**) is as compared to the retardance of a planar sample in the absence of voltage. Square wave voltage 5.31 V in (**a**–**d**) and 8.00 V in (**e**), $f = 200$ kHz; 105 °C.

hydrodynamic flows around these stationary deformations. The schemes are not to scale, as the in-plane period is about two orders of magnitude larger than the cell thickness $d$. All patterns involve polarization-related charges, either at the surface when **P** is tilted, of a surface density $\sigma_s = P\psi$, or in the bulk, when a splay deformation creates a bound charge density $\rho_b = -\mathrm{div}\mathbf{P}$. The question is whether and how these charges might be screened.

**In the homogeneous oscillations case**

$U < U_{ST}$, the surface charge $\sigma_s = P\psi_1 \approx 3 \times 10^{-4}\,\mathrm{C\,m^{-2}}$ is caused by the tilt of **P**; here $P = 6 \times 10^{-2}\,\mathrm{C\,m^{-2}}$ and $\psi_1 = 5 \times 10^{-3}$ rad. This charge oscillates with the same frequency as the applied field. An interesting point is that these oscillations are shifted in phase by almost $\pi/2$ with respect to the electric field inside the cell and the current $I_1$ through it. According to Eq. (9), the impedance of the $N_F$ slab is almost real, as the imaginary part is small if the frequency is less than about 1 MHz. Thus, the voltage $U_1$ acting in the slab and the current $I_1$ through it are practically in phase. On the other hand, Eq. (6) demonstrates that the tilt $\psi_1$ of polarization has a phase shift by about $\pi/2$ from $I_1$ and the

electric field $E_1$. Therefore, when the instantaneous tilt is zero, the electric field is at its extremum and vice versa. This feature underscores that the oscillatory regime is controlled by the viscous torques as opposed to the elastic torques. In the elasticity-driven phenomenon, an extremum of the field would cause an extremum of tilt and extremum of surface charge.

Free ions do not affect the dynamic patterns at the high frequencies of the field explored in this work. In the denominator of Eq. (6), the first term $P^2\cos^2\bar{\psi} \approx P^2$ is much larger than the imaginary part of the second term, which writes $\gamma\sigma_\perp$; with the rotational viscosity $\gamma = 5$ Pa•s [11] and $\sigma_\perp \sim 10^{-7}$ S/m, one estimates $\frac{P^2}{\gamma\sigma_\perp} \sim 10^4$, which supports the statement that the free ions do not play a role in the oscillatory regime.

To summarize, the $N_F$ cell's response to the weak ac field is through the oscillations of **P** with the frequency of the applied field. There is no voltage threshold for the occurrence of oscillations. As the voltage increases past a threshold $U_{ST}$, a new stripe structure emerges in which the oscillations are centered at the stationary splay and twist of **P**.

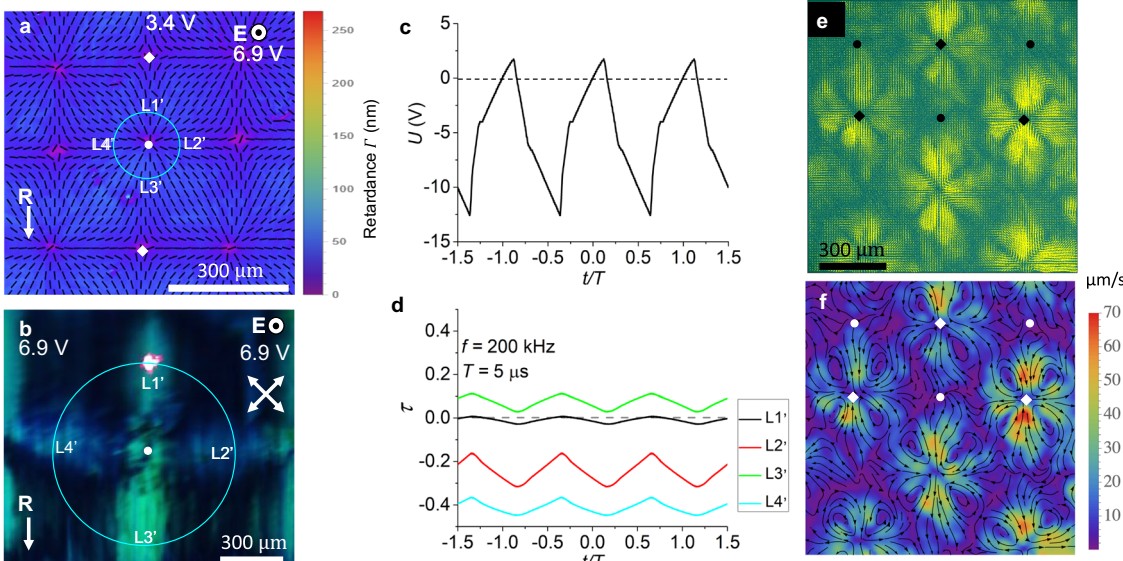

**Fig. 6 | Dynamics of splay-bend periodic Fréedericksz transition in a planar $N_F$ cell under a high voltage. a** PolScope Microimager texture of the in-plane splay and bend for the normal incidence of light; wavelength 535 nm. **b** Polarizing microscopy of square lattice of +1/−1 defects. **c** Potential difference measured at the $N_F$ cell electrodes; the generated voltage. **d** Transmitted intensity as a function of time at locations L1'–L4'. **P** oscillates with the frequency of the applied field. Incident laser beam makes an angle $\beta = 15°$ with the normal $\hat{z}$ to the cell. Dashed line corresponds to zero voltage. In (**a**–**d**), $d = (3.0 \pm 0.1)$ µm; applied voltage from the source $U_{rms} = 30$ V, 200 kHz sinusoidal waveform; 120 °C. **e** Particle image velocimetry (PIVlab, Matlab) integrated trajectories of fluorescent spherical flow tracers of the diameter 300 nm in the square lattice of +1/−1 defects; $d = (4.8 \pm 0.1)$ µm cell; sinusoidal wave of voltage 4.5 V and frequency $f = 200$ kHz; 115 °C. **f** the same cell, in-plane velocity field of the tracers.

## In the splay-twist pattern

The field-imposed stationary tilt $\bar{\psi}$ of **P** can create both the surface and bulk charges. The textures in Figs. 3, 4 indicate clearly that the optic axis and polarization tilt away from the rubbing direction **R** in the $xy$ plane by a stationary angle $\bar{\varphi}$, thus creating an in-plane splay $\frac{\partial P_y}{\partial y}$. The in-plane splay $\frac{\partial P_y}{\partial y}$ induces a bound charge which can be reduced if the polarization experiences deformation also along the $z$-axis. The bound charge $\rho_b = -\left(\frac{\partial P_y}{\partial y} + \frac{\partial P_z}{\partial z}\right)$ is reduced by the splay-cancellation principle when $\frac{\partial P_y}{\partial y}\frac{\partial P_z}{\partial z} < 0$, as in Figs. 3f and 7b, c. The analysis above and Figs. 3, 4 suggest the splay-twist pattern of **P**, comprised of splay, dominant at $y = 0, \pm L, ...,$ and twist, dominant at $y = \pm\frac{L}{2}, \pm\frac{3L}{2}, ...,$ where $L$ is the half-period. One can approximate the polarization field within the deformed layer of a characteristic extension $\lambda$ near the bottom plate in Fig. 7b, c as $\mathbf{P} = (P_x, P_y, P_z) \approx P(1, \bar{\varphi}_m \sin\frac{\pi y}{L}\cos\frac{\pi z}{2\lambda}, -\bar{\psi}\cos\frac{\pi y}{L}\sin\frac{\pi z}{2\lambda})$, where $\bar{\varphi}_m$ and $\bar{\psi}$ are the maximum azimuthal and polar tilts of **P**, respectively. The spatial derivatives along the $z$- and $y$- axes are of opposite signs, thus reducing the bound charge, $\rho_b = \frac{\pi P}{2}\left(\frac{\bar{\psi}}{\lambda} - \frac{2\bar{\varphi}_m}{L}\right)\cos\frac{\pi y}{L}\cos\frac{\pi z}{2\lambda}$, and the electrostatic energy density $f \propto \frac{1}{\varepsilon\varepsilon_0}[\mathrm{div}(\mathbf{P})]^2$. Integrating $f$ over the cross-section of the $\lambda$ layer yields the energy per unit length of a stripe, $F \propto \frac{\pi^2 P^2}{4L\lambda}(L\bar{\psi} - 2\lambda\bar{\varphi}_m)^2$, which is smaller than the homogeneous splay electrostatic energy $F_o \propto \frac{\pi^2 P^2 L}{4\lambda}\bar{\psi}^2$. The screening is most effective when $\frac{L}{2\lambda} = \frac{\bar{\varphi}_m}{\bar{\psi}}$. Since in the experiment $\frac{\bar{\varphi}_m}{\bar{\psi}} \sim 5$, we estimate $\frac{L}{\lambda} \sim 10$, which implies that the characteristic scale of deformations is larger than the cell thickness $d$. In the experiment, a typical value of $L$ is 200 µm, Fig. 3, thus $\lambda \approx 20$ µm.

If there were no splay cancellation, the bound charge density could be estimated as $\left|\frac{\partial P_z}{\partial z}\right| \sim P\frac{\bar{\psi}}{\lambda} \sim 3 \times 10^2$ C m$^{-3}$, where $\bar{\psi} \approx 0.1$ rad. Some of it might be screened by the free charges of ions, always present in liquid crystals, of a density $\rho_f \approx en$, where $e = 1.6 \times 10^{-19}$ C is the

elementary charge and $n$ is the concentration of ions. Independent measurements by the field reversal technique[24,25] yield $n \approx 2.4 \times 10^{20}$ m$^{-3}$ in the N phase (135 °C), which is in the typical range[26]. The corresponding free charge density $\rho_f \approx en \approx 50$ C m$^{-3}$ is noticeably smaller than $\left|\frac{\partial P_z}{\partial z}\right|$. Of course, $n$ can increase in the strongly polar environment of $N_F$, for example, through dissociation of some molecules and absorption from the surroundings. However, the existence of stationary splay in the horizontal plane with a sign opposite to the stationary splay along the $z$-axis demonstrates that the ionic screening could only be partial and that the splay-canceling emerges as a geometrical means to reduce $\rho_b$.

To understand which facet of the $N_F$-electric field coupling causes the stationary splay-twist deformation, let us consider the balance of the torques for a stationary tilt $\bar{\psi}$ in the $xy$-homogeneous state described above:

$$P(E_1\Psi_{-1} + E_{-1}\Psi_1)\sin\bar{\psi} + \varepsilon_0\Delta\varepsilon E_1 E_{-1}\cos\bar{\psi}\sin\bar{\psi} + K_{eff}(\bar{\psi})\bar{\psi}'' = 0, \quad (10)$$

where the variables with the subscript "−1" are the complex conjugates of the first-harmonic variables with the subscripts "1". According to Eq. (2), the phase shift between $\Psi_1$ and $E_1$ is very close to $\pi/2$, thus the first term $\propto P$ describing the polarization torque in Eq. (10) is negligibly small. Therefore, the stationary tilt is defined by the balance of the dielectric and elastic terms in Eq. (10), which implies that the periodic splay-twist deformations are initiated by the dielectric coupling of the $N_F$ with the applied electric field. The dielectric-elasticity balance is similar to that one in the conventional Fréedericksz transition in an N. When $|E_1| \geq \frac{\pi}{d}\sqrt{\frac{K_{11}}{\varepsilon_0\Delta\varepsilon}}$, the homogenous state can become unstable and develop stationary tilts. The important difference between the homogeneous Fréedericksz transition in the N and the periodic Fréedericksz in the $N_F$ is that the induced deformations in the $N_F$ are reshaped by the requirement to reduce the space charge, which makes the geometry spatially modulated in the $xy$ plane orthogonal to the field rather than homogeneous as in a conventional Fréedericksz transition in the N.

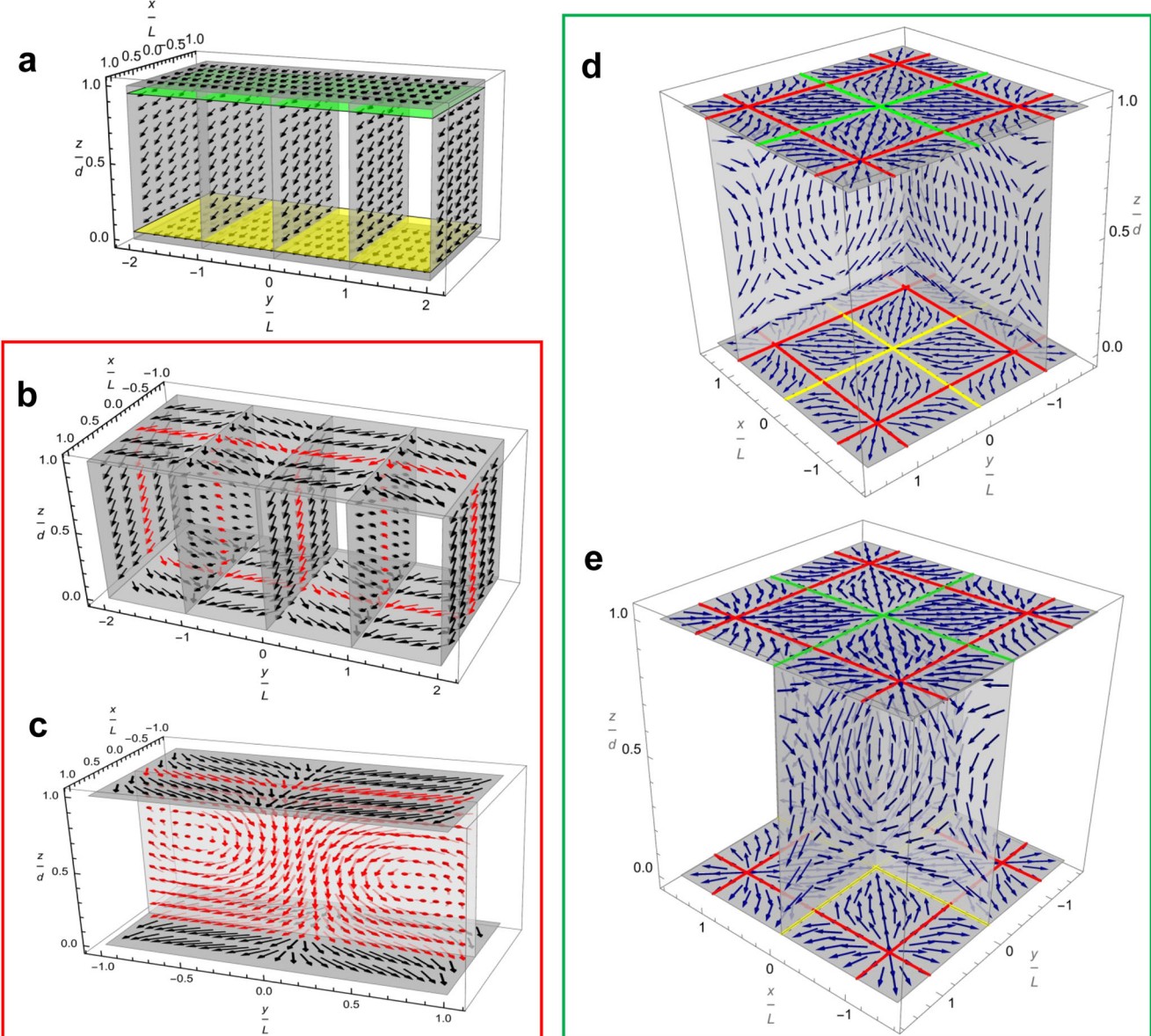

**Fig. 7 | Polarization patterns in N$_F$ splay Fréedericksz transition.**
**a** Homogeneous tilt of **P** with a positive bound charge near the bottom and negative charge near the top. **b**, **c** Two views of the splay-twist stripe pattern; in the $xz$ planes, tilts of **P** alternate from one S-region to the next; the splay of **P** in the horizontal $z = 0, d$ planes is opposite in sign to the splay in the adjacent vertical plane thus decreasing the bound charge; $yz$ plane shows both twist (predominant at $z \to d/2, y = \pm \frac{L}{2}, \pm \frac{3L}{2}, ...$) and splay (predominant at $z \to 0, d, y = 0, \pm L, ...$). The polarization field at $x = 0$ is shown in red to visualize the splay cancellation in (**b**) and the complementary splay-twist deformations in (**c**). **d, e** Two views of the splay-bend lattice of +1/−1 defects; splay cancellation is effective in the areas with red lines; in the areas with yellow and green lines, the bound charge is positive and negative, respectively.

Splay cancellation has been previously discussed for the non-polar director $\hat{\mathbf{n}}$ in a hybrid aligned N[22,27,28] and suspended N[29] films and for periodic Fréedericksz transition in a polymer N with a large $K_{11}$ [30,31]. The patterns in Figs. 3–6 with a large period, $2L \gg d$, are different from the patterns in a polymer N in which $2L \approx d$ [30]. The large $2L/d$ can be caused by the large $\bar{\varphi}_m / \bar{\psi}$ ratio. Another principal difference is that in the N, the splay cancellation is the elastic effect, which reduces the splay elastic energy density $\frac{1}{2} K_{11} \left( \text{div} \hat{\mathbf{n}} \right)^2$, while in the N$_F$, it is an electrostatic effect that reduces the total divergence div **P** of the polar vector. Electrostatic splay cancellation was suggested also for disclinations in a chiral N$_F$ [32], in which the externally imposed splay of a Grandjean-Cano wedge is relaxed by the in-plane splay of an opposite sign. A similar electrostatic splay cancellation was demonstrated for domain walls in a uniaxial solid ferroelectric[33] that adopt a saddle-point morphology to reduce the space charge between the domains with antiparallel polarization.

The bound charge can form as a result of flexoelectric polarization, $\rho_f = -\text{div} \mathbf{P}_f$, where $\mathbf{P}_f = e_1 \hat{\mathbf{n}} \, \text{div} \hat{\mathbf{n}} - e_3 \, \hat{\mathbf{n}} \times \text{curl} \hat{\mathbf{n}}$, $e_1$ and $e_3$ are flexoelectric coefficients of splay and bend, respectively[6]. Near the bottom plate and in the S-regions, $z = y = 0$, the flexoelectric splay charge is maximum $\rho_f = -\frac{\pi^2}{4} e_1 \left( \frac{\bar{\psi}}{\lambda} - \frac{2\bar{\varphi}_m}{L} \right)^2$. For the typical $e_1 = 10^{-11}$ C/m (and even for a few orders of magnitude larger $e_1$, ref. 34), $P = 6 \times 10^{-2}$ C m$^{-2}$ and experimentally observed $L \approx 3 \times 10^{-4}$ m, one concludes that the flexoelectric charge in the explored patterns is smaller than the polarization charge, $\left| \rho_f / \rho_b \right| = \left| \frac{\pi}{2P} e_1 \left( \frac{\bar{\psi}}{\lambda} - \frac{2\bar{\varphi}_m}{L} \right) \right| \leq \left| \frac{\pi \bar{\varphi}_m}{LP} e_1 \right| \sim 10^{-5}$.

Whenever the surface or bulk stationary charges in Fig. 7b–e are not fully compensated by the splay cancellation and ions, they create an in-plane electric field. For example, on the top plate in Fig. 7b, the electric field caused by the horizontal splay of **P** would be directed from the stripe $y/L = 0$ to the stripes $y/L = -1$ and $y/L = 1$. These fields are opposite to local directions of **P**. The phenomenon is rooted in the electrostatics of the problem: If the polarization and bound charge-induced field were parallel, then the electrostatic energy will be negative, and the system would become spatially nonuniform even at the vanishingly small voltages.

To summarize, the splay-twist stripes are caused by the balance of dielectric and elastic torques. The field-induced splay in the vertical cross-section of the cell is reduced by the in-plane splay of an opposite polarity. This electrostatic splay cancellation produces patterns with a period much larger than the thickness of the cell. As the voltage increases further, the one-dimensional deformations in the plane of the cell are replaced by two-dimensional deformations with a square lattice of $-1$ and $+1$ defects; the latter are of splay geometry.

### The lattice of $+1/-1$ defects

Offers similar considerations of dielectric-elasticity balance and splay cancellation. At higher voltages, $U > U_{SB}$, a dramatically reduced retardance suggests that **P** is strongly tilted towards **E**, which replaces twist with bend, Fig. 7d, e and Supplementary Figs. 9, 10. In Fig. 7d, e, **P** is along **E** in the middle of the cell and experiences splay and bend near the plates. The splay cancellation is clearly seen at intersections of horizontal and vertical planes marked by red, at which the in-plane derivatives $\frac{\partial P_x}{\partial x}$ and $\frac{\partial P_y}{\partial y}$ are of opposite signs as compared to $\frac{\partial P_z}{\partial z}$ in the vertical cross-sections. For example, the polarization in the vicinity $R \ll L$ of the radial $+1$ defects at the bottom plate in Fig. 7d, e with the cores at the intersection of red lines can be modeled as $\mathbf{P} = \left(P_{\bar{x}}, P_{\bar{y}}, P_z\right) \approx P\left(\frac{\bar{x}}{R}\cos\frac{\pi z}{\lambda}, \frac{\bar{y}}{R}\cos\frac{\pi z}{\lambda}, -\sin\frac{\pi z}{2\lambda}\right)$; here the origin of coordinates $(\bar{x}, \bar{y}, z)$ is at the core of each $+1$ defect. The bound charge is then $\rho_b = P\left(\frac{\pi}{2\lambda} - \frac{1}{R}\right)\cos\frac{\pi z}{\xi}$, being smaller than the charge $\rho_b = \frac{\pi}{2\lambda}P\cos\frac{\pi z}{\lambda}$ of a one-dimensional splay $\mathbf{P} \approx P\left(\cos\frac{\pi z}{\lambda}, 0, -\sin\frac{\pi z}{2\lambda}\right)$. However, there are also regions in which the bound charge is not splay-canceled because $\frac{\partial P_y}{\partial y}$ and $\frac{\partial P_z}{\partial z}$ are of the same sign. For example, **P** converges towards the yellow lines at the bottom of the cell, $z = 0$, and diverges away from the green lines at the top plate, $z = d$. As a result, the bound charge along these lines and in the adjacent regions are positive and negative, respectively, $\rho_b = \pm P\left(\frac{\pi}{2\lambda} + \frac{1}{R}\right)$. In the vicinity of $-1$ defect cores at the intersection of red and yellow lines or red and green lines, splay cancellation is observed only along one of the directions, $\rho_b = P\left(\frac{\pi}{2\lambda} \pm \frac{x^2}{R^3} \mp \frac{y^2}{R^3}\right)$. Importantly, the positive and negative bound charges are facing each other across the small separation $d$ in Fig. 7d, e; in an alternative scheme, Supplementary Fig. 11, bound charges of opposite signs are separated by much larger distances $\sim L$, which implies a higher electrostatic energy. The symmetry of the polarization field in the vertical cross-section of the cells in Fig. 7d, e is supported by an experimental observation that the interference colors of the cell tilted by $45°$ around the $y$- or $x$-axis are the same within the quadrants of the elementary cell, Supplementary Fig. 10c. The appearance of potential difference at the electrodes of the $N_F$ cell with the square lattice, Fig. 6c, supports the scheme in Fig. 7d, e. In the homogeneous oscillatory state, this stationary potential difference is zero, as expected. We associate this potential difference in Fig. 6c with the bound charges in the square lattice in Fig. 7d, e; this potential difference is the apparent reason for hydrodynamic flows in the square lattices, Fig. 6e, f.

The square lattice thus exhibits simultaneously numerous mechanisms of coupling of the electric field to the liquid crystal structure: oscillations of polarization, dielectric-elastic balance that produces stationary splay and bend, accumulation of stationary potential differences at the electrodes of the cell as a result of deformations produced by a high-frequency ac voltage with zero dc component, and, finally, occurrence of hydrodynamic flows.

To summarize, a planar $N_F$ cell activated by a high frequency transversal electric field shows a hierarchy of static and dynamic responses. At low voltages, the electro-optic response appears as a thresholdless homogeneous oscillations of polarization **P**, followed at higher voltages by stationary periodic splay-twist and splay-bend of **P**. The polarization oscillates with the frequency of the applied field in all these regimes, with the difference that in the splay-twist and splay-bend states the oscillations are centered at the stationary deformations. The amplitude of high-frequency oscillations is controlled by the balance of electric and viscous torques, which is different from the conventional nematic Fréedericksz transition in which the stationary director deformations are controlled by the balance of electric and elastic torques.

The appearance of the splay-twist and splay-bend states is counterintuitive since in the absence of the electric field the polarization field prefers to form structures with bend (such as circular domains[18–20]) and twist[15] but not splay since it creates bound charges. Splay deformations are predominant in the described periodic splay $N_F$ Fréedericksz transition. For example, in the $+1/-1$ lattices, Fig. 5, all $+1$ defects are of a radial splay structures while in the field-free states, $+1$ defects prefer to be of a circular (bend) type[18–20]. The onset of these stationary deformations is caused by the balance of the dielectric and elastic torques. In the splay-twist stripes, the $N_F$ finds a geometrical splay-canceling solution to reduce the total splay div **P** and the associated bound charge: the splay in the vertical plane $\frac{\partial P_z}{\partial z}$ imposed by the external electric field is reduced by the splay $\frac{\partial P_y}{\partial y}$ of an opposite polarity, so that $\frac{\partial P_y}{\partial y} + \frac{\partial P_z}{\partial z} \to 0$. In the splay-bend square lattice, the splay cancellation leaves regions of finite bound charges which produce a stationary potential difference at the cell's electrodes. This potential difference is another unexpected facet of the $N_F$ response to the high-frequency unbiased excitation; it is an apparent cause of the onset of electrohydrodynamic flows in the splay-bend textures.

The seemingly simple setting of the splay Freedericksz effect in the $N_F$ shows nontrivial interplay of electrostatics and geometry, high frequency excitations and stationary deformations, ac-induced dc potential differences and electrohydrodynamic flows. One should expect that the geometrical splay-canceling of electrostatic charges should be operational in other geometries when external factors, not necessarily the electric field, such as confinement, impose a splay. One example has been recently presented for a chiral $N_F$ in a Grandjean-Cano wedge, in which the disclination line adopted a zig-zag shape to balance the wedge-induced splay with the splay of an opposite polarity in the plane of the cell[32]. We expect more examples of geometry-electrostatics interplay in fluid ferroelectrics such as $N_F$ and the recently discovered twist-bend ferroelectric nematics[35] to follow, since deformations of polarization in these materials are free of limitations imposed by crystallographic directions or density modulations.

## Methods

### Substrate preparation

A PI2555 alignment layer of a thickness 50 nm is spin-coated onto the lithographically patterned ITO-glass substrates following ref. 13. The sheet resistance of the ITO electrodes is $10\,\Omega/\square$. The thickness of alignment layer is measured by Digital Holographic Microscopy. The PI2555 layer is unidirectionally rubbed using a Rayon YA-19-R cloth (Yoshikawa Chemical Company, Ltd, Japan) of a thickness 1.8 mm and filament density 280/mm². An aluminum brick of a length of 25.5 cm, width of 10.4 cm, height of 1.8 cm, and weight of 1.3 kg, covered with the cloth, imposes a pressure of 490 Pa at a substrate and is moved ten times with a speed of 5 cm/s over the substrate; the rubbing length is about 1 m [13]. The buffed polyimide aligns the N director parallel to the buffing direction with a small pretilt $\approx 3° \pm 1°$ [36]. In the $N_F$ phase of RM734, **P** is

along the rubbing direction **R**, as verified by applying an in-plane electric field to a planar cell with two PI2555 plates assembled with parallel orientation of the rubbing directions. These experiments also suggest that the pretilt of **P** at the PI2555 substrates is negligibly small since the optical retardance $\Gamma$ equals $(n_e - n_o)d$, and does not change when the field is applied.

## Synthesis of RM734

RM734 was synthesized by Enamine (https://enamine.net/) as described in ref. [37] and purified by silica gel chromatography and recrystallization in ethanol.

## Cell assembly

The cells are formed by two ITO- and PI2555-covered glass substrates. The rubbing directions on the two plates are parallel to each other. The electrode area is $1\,cm^2$. The cell gap thickness $d$ is set by glass spherical spacers in the range 2–10 μm; $d$ is measured at five different locations within the cell by an interferometric technique using UV/VIS spectrometer Lambda 18 (Perkin Elmer). The standard variation of $d$ is 0.1 μm or less. The liquid crystal is filled into cells in the isotropic phase. The temperature is controlled by a Linkam hot stage with the accuracy $\pm 0.1\,°C$.

## Refractive indices measurements

The ordinary $n_o$ and extraordinary $n_e$ refractive indices for the N and $N_F$ phases of RM734 are determined using a wedge cell as described in ref. [15] with an accuracy of 0.5%.

## Response to electric field

A vertical electric field $\mathbf{E} = (0, 0, \pm E)$ is applied to the ITO electrodes at the top and bottom plates. A Siglent SDG1032X waveform generator and an amplifier (Krohn-Hite corporation) generate the square and sine ac field of frequencies between 0 Hz and 2 MHz. The voltages such as $U_{ST}$ and $U_{SB}$ are measured as rms values of the potential differences across the cell. In the caption to Fig. 6, $U_{rms}$ refers to the voltage generated by the source.

## Measurements of concentration of ions

A cell of a thickness $d = (20.0 \pm 0.1)$ μm and an electrode area $\Sigma = 1\,cm^2$ is connected to the 10 kΩ resister in series, Supplementary Fig. 12. A square waveform of 0.5 Hz and 3.5 V is applied by using the Siglent SDG1032X waveform generator. The current through the resistor is measured with an oscilloscope Tektronix TDS 2014 (sampling rate 1GSa/s). When the polarity of the square wave is switched, this mobile ion current adds to the discharge current[24,25]. To calculate the ion concentration from the current bump induced by the mobile ions in liquid crystal: ion concentration $n = \frac{1}{e\Sigma d}\int_0^t I(t)dt$, where $\int_0^t I(t)dt$ is the area of the current bump produced by field reversal[24,25].

## Polarizing optical microscopy observation of textures

A polarizing optical microscopes Nikon Optiphot-2 equipped with a QImaging camera and Olympus BX51 with an Amscope camera were used for observations of textures. The director field and optical retardance were mapped by PolScope MicroImager. A full-waveplate 550 nm optical compensator was used to verify the director orientation. Analysis of optical activity was performed in monochromatic light with blue and green interferometric filters (center wavelength $\lambda = 485$ nm and 532 nm, respectively, a bandwidth of 1 nm).

**Optical retardance of $N_F$ cell for oblique incidence of light**. We consider an electromagnetic wave propagating through an $N_F$ slab between two glass plates. The dielectric tensor at optical frequencies (the optic tensor) $\boldsymbol{\varepsilon}$ of the $N_F$ oscillates in the $xz$ incidence plane. For a monochromatic wave $\propto e^{-i\omega t}$, the homogeneity of the structure in the $xy$

plane preserves the in-plane wave vector $\mathbf{q} = \nu n_g \sin\beta_g \hat{\mathbf{x}}$, where $\hat{\mathbf{x}}$ is the unit vector along the $x$-axis, $\nu = \omega/c = 2\pi/\lambda$ is the free space wavenumber, $n_g$ is the refractive index of the glass plate, and $\beta_g$ is the incidence angle in the glass plate calculated from the $z$-axis, which is very close to the 'ordinary' angle of light incidence $\beta = \arcsin\left(\frac{n_g \sin\beta_g}{n_o}\right)$. The instantaneous phase retardance is $\Gamma(\psi) = \left(k_x - k_y\right)d$, where $k_y = n_o\nu\cos\beta$ and

$$k_x = \frac{n_o\nu}{1 + 2\delta\sin^2\psi}\left[\delta\sin\beta\sin 2\psi + \sqrt{(1 + 2\delta)\left(\cos^2\beta + 2\delta\sin^2\psi\right)}\right] \quad (11)$$

are normal ($z$) components of the wavevectors of the ordinary and extraordinary waves that are polarized in the $xz$ plane and along the $y$-axis, respectively. In the resulting expression for $\Gamma(\psi)$, Eq. (1), the first and the second terms are odd and even functions of $\beta$ and $\psi$, respectively. This feature allows one to separate their contributions and to determine the dependence of the tilt angle on the applied voltage.

## Fluorescence confocal polarizing microscopy

A fluorescent dye $n,n'$-bis(2,5-di-tert-butylphenyl)-3,4,9,10-perylenedicarboximide (BTBP) is added in tiny quantities of 0.02% (by weight) to the liquid crystal RM734. The transition dipole of BTBP aligns parallel to the molecular director of RM734. Polarization of the excitation beam (488 nm, Ar laser) is set by the linear polarizer **P** in the Olympus Fluoview BX-50 confocal microscope. The fluorescent signal passes through the same polarizer. The intensity of fluorescence is maximum when **P** ∥ **R** and minimum when **P** ⊥ **R**.

## Particle image velocimetry

A small amount (0.5 wt%) of fluorescent silicon oxide spheres of diameter 300 nm (Corpuscular, Inc.) are added to the liquid crystal material. Their trajectories are uncovered by particle image velocimetry package PIVLab[38] in Matlab.

## Numerical simulations of twist

Numerical simulations are performed as described in ref. [15] using $n_o = 1.52$ and $n_e = 1.79$ refractive indices measured at 120 °C for a cell of a thickness 3.1 μm.

## Data availability

All data needed to evaluate the conclusions in the paper are present in the article and in the Supplementary Figures. Source data are provided with this paper.

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

## Acknowledgements

The authors thank Dr. Hari Krishna Bisoyi and Organic Synthesis Facility at the AMLCI for the purification of RM734, and P. Kumari and M.O. Lavrentovich for useful discussions. This work was supported by NSF grants ECCS-2122399 (O.D.L., S.V.S.; electro-optic experiments) and DMR-2341830 (O.D.L., modeling), NASU project No. 0121U109816 (V.G.N., O.K., O.B., N.A.), and the Long-term program of support of the Ukrainian research teams at the Polish Academy of Sciences carried out in collaboration with the U.S. National Academy of Sciences with the financial support of external partners via the agreement No. PAN.BFB.S.BWZ.356.022.2023 (V.G.N., O.K.). The authors also acknowledge funding from the NATO SPS project G6030 (V.G.N., O.D.L.).

## Author contributions

V.G.N. initiated the experiments. B.B., O.K., O.B., V.G.N., and S.P. performed the experiments. N.A., S.V.S., and O.D.L. developed the models. B.B., S.P., S.V.S., and O.D.L. analyzed the experimental data. B.B., S.V.S., S.P., and O.D.L. wrote the manuscript with input from all co-authors. O.D.L. and V.G.N. supervised the project. All authors have read and agreed to the submitted version of the manuscript.

## Competing interests

The authors declare no competing interests.

## Ethics

Research has been conducted at Kent State University and Institute of Physics, National Academy of Sciences of Ukraine; all contributors have been acknowledged.
