## [Peer Review file · Nature Communications]

Periodic splay Fréedericksz transitions in a ferroelectric nematic

Corresponding Author: Professor Oleg Lavrentovich

Version 0:

Reviewer comments:

Reviewer #1

(Remarks to the Author)

The present paper ("Periodic splay Frederiks transitions in a ferroelectric nematic") by Lavrentovich et al makes a significant contribution to understanding the behaviour of ferroelectric nematics under applied fields. Interest in the topic of polar order in liquid crystals continues to grow, and this paper is surely deserving of publication in Nature Communications.

Minor comments

- * The authors report the synthesis of RM734 was done as per ref 5, however this paper simply repeats the originally reported synthesis from 10.1002/chem.201702742 (which is not cited)
- * Some figures would benefit from improvement; for example Fig 1b, 1c, 1f have rather small text.
- * Some subfigures are not referred to in the order they are presented. For ex., Fig 2c is mentioned first, then 2b. This isn't a major issue, but it does suggest a need to reorganize the text or the figure layout.
- * The authors refer to "electrohydrodynamic flows" on line 129; this might be a good opportunity to use video as an additional supplementary resource.

Reviewer #2

(Remarks to the Author)

The submitted paper by Basnet et. al. covers the dynamic POM textures and optical defects formed when an ac field is applied to a ferroelectric nematic liquid crystal. This is a fundamental technique used to characterise liquid crystals, and has been used in the display industries to great effect, and so understanding the behaviour of the phase in this experiment is a significant step towards development of applications utilising the optics of this phase.

The analysis of the data and subsequent suggested mechanism of splay cancellation is both thorough and convincing leading to an extremely clear presentation of some beautiful optical textures and observations. The methodology is sound and well documented. I do have several questions and comments, but these are in general minor and do not represent a significant barrier to publication. I feel the weakest element of the paper is the "flow" as the layout of the figures and the way they are referenced in the text perhaps could be improved on to add readability.

Line 42 - Initially yes its splay but definitely bend occurs too or else how do we measure k_1 and k_3 from CV curves.

Line 42- This reference does not support this. In fact they show the opposite effect stating "It should be noted that the electric switching current in the NF phase is accompanied by clear optical switching, director orientation switches between splay state at zero applied voltage and uniform states along applied field". It is true for the N_x phase however.

General intro – It should probably be specified for the general reader who may be less familiar with the Nf phase that P lies along the long axis and director particularly leading into paragraph 3

General question – A limitation of CV Frederiks curve analysis is being unable to measure K_1 K_2 and K_3 simultaneously

needing different alignments and electrode patterns. This suggested mechanism requires all three splay twist and bend deformations so hypothetically could you obtain measurements of all three coefficients by analysing the single CV curve (or retardance V curves)

Line 63 – Please define all terms eg. What is d . I know its all in extended data figures but should be clear in the main.

Line 74 – Obviously we never get close since the reorientation mechanism changes but is there some significance of U_{st} .

Line 101 – Rotated in main but tilted in esi. Pick one and stick to it!

Line 128 – Please include verification of these textures and patterns in the ESI.

Line 152 – Tilt value needs units. assume its radians?

Line 155 – Is that assumption valid? Could the free charge coordinate to the polarisation in some way that would effectively render it “bound”? Though this just exacerbates the lack of ions to mask the actual bound charge

Line 158 – How? Molecular degradation? Certainly possible but I would like some suggestions.

Line 159 – Perhaps make it clear that you are explicitly suggesting that free charge can compensate for the $\text{div}P$ below U_{st} .

Could you give a rough estimate for the size of ξ

Line 191 – Is a typical e_1 likely representative of the NF phase considering the N phase for these materials has been suggested to be “colossal” [10.1103/PhysRevMaterials.7.035603].

Line 213 – Wouldn't this be distinguishable by the same twist analysis presented earlier?

It feels like all three discussion sections could benefit from a single sentence to at the end summarising the competing effects and which one takes over from which one leading to each pattern observed.

Figures 1 and 2 feel like they could be arranged more clearly to allow for an easier reading experience for the reader. It very much feels like you have to bounce all over the figure and leading to the feeling that they are not called in the correct order in the text (i.e. 1b before 1a). I don't believe they are called in an incorrect order in the text but the sensation of having to continuously look back at previous figures makes the paper more complicated to follow.

Reviewer #4

(Remarks to the Author)

Frederiks transition in a ferroelectric nematic liquid crystalline phase (NF) is reported under the influence of an ac electric field. In ferroelectric nematics, no Frederiks transition is observed if a dc field is applied to planar cells filled with liquid crystal in the NF phase, because polarization in the NF phase is so high that the external dc field can always be compensated by a rotation of polarization (which means rotation of the average direction of the nematic director, because polarization P is along the local average direction of the long molecular axes). On the other hand, when an ac field is applied and the amplitude is increased, authors observe three behavioural regimes: a homogeneous rotation at low amplitudes, periodic stripes of splay and twist deformation of P , and, at high enough amplitudes, periodic square lattice with splay and bend of P . The paper offers a thorough identification of the NF structure in all three regimes by using several experimental techniques. However, it does not address the origin/driving force for the formation of such structures. We would like to point out, that similar texture changes were reported also in conventional nematics, where a variety of pattern morphologies may arise due to the flexoelectricity and electroconvection at high ac field frequency (such effects were extensively studied by the group of A. Buka, see for example *Liquid Crystals Reviews*, 4:2, 101-134 (2016), by N. Éber et al, and also the paper by Sasaki et al in *Nat. Commun.* in 2016; DOI: 10.1038/ncomms13238). How and why were these mechanisms excluded? For conventional nematics, it was demonstrated that the AC voltage induces periodic patterns for example due to the ions accumulated at an electrically insulating polymer interface. Such one-dimensional and two-dimensional arrays of defects are particularly common for nematic phase with an ionic impurity. In the NF phase, as pointed out also by authors of the present manuscript, the ionic impurities are quite high, so self-organization of arrays of defects is quite probable.

Based on the above, we cannot recommend the acceptance of the paper in this form. However, if the authors can include in the manuscript a plausible explanation on the difference between the effect reported in conventional nematics and the effects leading to the formation of complex patterns that they observed in ferroelectric nematics, the paper can be considered for *Nature Communications*.

We also have some comments regarding the text and figures:

Text:

Line 63: The threshold voltage depends on the cell thickness, so we suggest that the cell thickness is given in the text as well (not only in the figure caption).

Line 196: titled -> tilted

Figures:

Figures are very well prepared and rich in information. However, we suggest that the authors consider the following comments:

- 1) The size of the text on the axes should be increased in all figures both in the main paper and the extended data.
- 2) For several figures, the symbols used in the figures are not introduced in the figure caption.

Additional comments to specific figures:

- 1) Figure 1: different structures are observed in the same cell just by increasing the voltage amplitude. What is the reason for the figure parts being given for three different cells of different thickness? Also, explain in the caption what are P and R in the figure.
- 2) Extended Data Figure 5: amplitude of the field -> amplitude of the voltage (U)

Version 1:

Reviewer comments:

Reviewer #1

(Remarks to the Author)

The revised paper is sound and suitable for publication in its current format, with a few final comments from us below.

Line 121-124. There are lots of inline equations in this section which I found harder to follow with them being inline. Possibly consider inputting them as display equations instead. Though this is just a suggestion for readability.

Line 125 could you state the "such a way" the compensators were used to aid in repeatability. Presumably both Γ and Γ_0 are related to the sum of both the retardance of the cell and the compensator here? Or how then does $2\pi/\lambda \cdot \Gamma_0 = \pi/2$?

Fig 2b,c clearly show some phase shift between the applied voltage and optical response. Fig4b,c shows maybe a smaller phase shift or maybe none at all? It is hard to tell from the figure and maybe a vertical line as in fig2 could be used here.

Would the considerations to the phase difference from fig2 apply equally to fig 4?

Fig.c. remark the "c" so it is not obscured. I also feel it would be useful to plot what the voltage profile applied by the generator is to show how distinct this change is.

Reviewer #2

(Remarks to the Author)

Reviewer #4

(Remarks to the Author)

The authors made extensive improvements of the manuscript regarding the clarity and they also included additional measurements to support their findings. All the points that were raised in the first report were considered. We thus recommend the paper to be accepted for publication.

We are thankful to the Reviewers for the assessment of our manuscript. We find their comments very useful for the improvement of the presentation and address all of them in the revised manuscript. We also performed extensive additional experiments and developed theoretical models to substantiate more fully our findings. Because of the additional research, we added three co-authors, Drs. Sathyanarayana Paladugu, Natelie Aryasova, and Sergij V. Shiyonovskii. We explored electro-optical response with submicrosecond resolution, in addition to the original stationary experiments. In the revised manuscript, we highlighted in yellow all the modifications in response to the Reviewer's comments. We list these changes below and hope that the modified manuscript answers all the Reviewers comments.

Replies to the Reviewers comments:

Reviewer #1

R1.1. The present paper ("Periodic splay Frederiks transitions in a ferroelectric nematic") by Lavrentovich et al makes a significant contribution to understanding the behaviour of ferroelectric nematics under applied fields. Interest in the topic of polar order in liquid crystals continues to grow, and this paper is surely deserving of publication in Nature Communications.

A1.1. We thank the Reviewer for the assessment of the manuscript and suggestions.

R1.2. Minor comments

* The authors report the synthesis of RM734 was done as per ref 5, however this paper simply repeats the originally reported synthesis from 10.1002/chem.201702742 (which is not cited)

A1.2. We corrected the omission and cited the original paper by Mandle et al (2017).

R1.3.* Some figures would benefit from improvement; for example Fig 1b, 1c, 1f have rather small text.

A1.3. We enlarged the texts.

R1.4.* Some subfigures are not referred to in the order they are presented. For ex., Fig 2c is mentioned first, then 2b. This isn't a major issue, but it does suggest a need to reorganize the text or the figure layout.

A1.4. We address this by re-organizing the figures, moving some plots from Supplement and adding new data.

R1.5.* The authors refer to "electrohydrodynamic flows" on line 129; this might be a good opportunity to use video as an additional supplementary resource.

A1.4. Thank you, we added three videos as well as the new Fig.6e,f.

Reviewer #2:

R2.1. The submitted paper by Basnet et. al. covers the dynamic POM textures and optical defects formed when an ac field is applied to a ferroelectric nematic liquid crystal. This is a fundamental technique used to characterise liquid crystals, and has been used in the display industries to great effect, and so understanding the behaviour of the phase in this experiment is a significant step towards development of applications utilising the optics of this phase.

The analysis of the data and subsequent suggested mechanism of splay cancellation is both thorough and convincing leading to an extremely clear presentation of some beautiful optical textures and observations. The methodology is sound and well documented. I do have several questions and comments, but these are in general minor and do not represent a significant barrier to publication.

A2.1. We thank the Reviewer for the assessment of the manuscript and suggestions to improve it.

R2.2. I feel the weakest element of the paper is the “flow” as the layout of the figures and the way they are referenced in the text perhaps could be improved on to add readability.

A2.2. We agree and have revised the manuscript substantially, adding theoretical analysis and experiments on the dynamics of the effect and streamlining the flow.

R2.3.Line 42 - Initially yes its splay but definitely bend occurs too or else how do we measure k_1 and k_3 from CV curves.

A2.3. Thank you, we mentioned bend on line 47.

R2.4. Line 42- This reference does not support this. In fact they show the opposite effect stating “It should be noted that the electric switching current in the NF phase is accompanied by clear optical switching, director orientation switches between splay state at zero applied voltage and uniform states along applied field”. It is true for the Nx phase however.

A2.4. Thank you, we were referring to the next sentence in the reference, which indicated a difficulty of realignment. We replaced the argument with the explicit experiment by Chen et al (2020), which compared directly the Freedericksz response of the N and NF phases in the same cell.

R2.5. General intro – It should probably be specified for the general reader who may be less familiar with the Nf phase that P lies along the long axis and director particularly leading into paragraph 3

A2.5. We added this clarification in the Introduction, lines 32-34.

R2.6. General question – A limitation of CV Frederiks curve analysis is being unable to measure K_1 , K_2 and K_3 simultaneously needing different alignments and electrode patterns. This suggested mechanism requires all three splay twist and bend deformations so hypothetically could you obtain measurements of all three coefficients by analysing the single CV curve (or retardance V curves)

R2.6. Extracting all three elastic constants from a single experiment would be a tremendous achievement for the science of the NF but we cannot do it yet because of the limitations of our knowledge. Even the measurements of dielectric permittivities represent a challenge, as follows from the current literature.

2.7. Line 63 – Please define all terms eg. What is d . I know its all in extended data figures but should be clear in the main.

A2.7. We added the definitions, line 82.

R2.8. Line 74 – Obviously we never get close since the reorientation mechanism changes but is there some significance of U_{sat} .

A2.8. We performed additional experiments with oblique incidence of light to measure the amplitude of polarization tilts and found that these tilts are oscillating in time and that their amplitude is less than 1 degree. The amplitude is proportional to the applied field, as shown in the added Fig.2d, but we cannot reach a state at which the amplitude saturates, thus we cannot say much about U_{sat} . We keep the original statement that the linear dependence of the tilt on the voltage is similar to the model predictions by Clark for the dc field; in that model U_{sat} is the limiting voltage that the realigned polarization could fully screen at the end of the section *(B) Uniformity of polarization tilt.*, lines 196-198 (we do not highlight this original statement):

“The model proposed by Clark et al. ¹ for “block reorientation” of \mathbf{P} by a dc field exhibit a similar z-independence of the tilt and its linear growth with the voltage, as in Fig.2d.”

I

R2.9. Line 101 – Rotated in main but tilted in esi. Pick one and stick to it!

A2.9. We changed rotated to tilted.

R2.10. Line 128 – Please include verification of these textures and patterns in the ESI.

A2.10. We added three videos and Fig.6e.f to illustrate electrohydrodynamics.

R2.11. Line 152 – Tilt value needs units. assume its radians?

A2.11. Yes, we added “rad”.

R2.12. Line 155 – Is that assumption valid? Could the free charge coordinate to the polarisation

in some way that would effectively render it “bound”? Though this just exacerbates the lack of ions to mask the actual bound charge

A2.12. Thank you for the comment. We do not need to make this assumption since the in-plane splay demonstrates that the bound charge is not screened by ions. We modified the text in the Discussion, section “*In the splay-twist pattern*”, lines 424-429:

“The corresponding free charge density $\rho_f \approx en \approx 50 \frac{\text{C}}{\text{m}^3}$ is noticeably smaller than $\left| \frac{\partial P_z}{\partial z} \right|$. Of course, n can increase in the strongly polar environment of N_F , for example, through dissociation of some molecules and absorption from the surroundings. However, the existence of stationary splay in the horizontal plane with a sign opposite to the stationary splay along the z -axis demonstrates that the ionic screening could only be partial and that the splay-cancelling emerges as a geometrical means to reduce ρ_b .”

R2.13. Line 158 – How? Molecular degradation? Certainly possible but I would like some suggestions.

R2.13. In the text cited above, we added, line 425-426, “for example, through dissociation of some molecules and absorption from the surroundings”

R2.14. Line 159 – Perhaps make it clear that you are explicitly suggesting that free charge can compensate for the $\text{div}P$ below U_{st} .

R2.11. To clarify the suggestion, we added the following text in the Discussion, section “*In the homogeneous oscillations case*”, lines 392-396:

“Free ions do not affect the dynamic patterns at the high frequencies of the field explored in this work. In the denominator of Eq. (6), the first term $P^2 \cos^2 \bar{\psi} \approx P^2$ is much larger than the imaginary part of the second term, which writes $\gamma \sigma_{\perp}$; with the rotational viscosity $\gamma = 5 \text{ Pa} \cdot \text{s}^2$ and $\sigma_{\perp} \sim 10^{-7} \text{ S/m}$, one estimates $\frac{P^2}{\gamma \sigma_{\perp}} \sim 10^4$, which supports the statement that the free ions do not play a role in the oscillatory regime.”

R2.15. Could you give a rough estimate for the size of ξ

A2.15. Yes, we added the following estimate and text after Eq. (6), lines 190-192

“This thickness can be estimated as $\xi = \sqrt{\frac{\epsilon_0 \epsilon K_{11}}{P^2}}$, where $\epsilon \sim 10 - 100$, Refs. ^{3,4}, $P = 6 \times 10^{-2} \frac{\text{C}}{\text{m}^2}$, Ref. ⁵, while K_{11} might be in the range 10-100 pN, Ref. ⁶. With these estimates, one finds $\xi = (0.5 - 5) \text{ nm}$, of a tiny molecular scale.”

R2.16. Line 191 – Is a typical e_1 likely representative of the Nf phase considering the N phase for these materials has been suggested to be “colossal” [10.1103/PhysRevMaterials.7.035603].

R2.16. We note that the flexocoefficient was measured as a ratio to the elastic constant in the cited paper and the said ratio was only two-three times larger than the corresponding ratio in a conventional nematic. These claims of “colossal” flexocoefficient do not change our conclusion, which we supplemented with a reference to the cited paper, lines 474-477:

“For the typical $e_1 = 10^{-11}$ C/m (and even for a few orders of magnitude larger e_1 , Ref. ⁷), $P = 6 \times 10^{-2} \frac{\text{C}}{\text{m}^2}$ and experimentally observed $L \approx 3 \times 10^{-4}$ m, one concludes that the flexoelectric charge in the explored patterns is smaller than the polarization charge, $[\rho_f/\rho_b] = \left| \frac{\pi}{2P} e_1 \left(\frac{\bar{\psi}}{\lambda} - \frac{2\bar{\varphi}_m}{L} \right) \right| \leq \left| \frac{\pi\bar{\varphi}_m}{LP} e_1 \right| \sim 10^{-5}$.

”

R2.17. Line 213 – Wouldn’t this be distinguishable by the same twist analysis presented earlier?

A2.17. We do not observe twists in the splay-bend square lattices, since the analysis with decrossed polarizers shows no optical activity.

R2.18. It feels like all three discussion sections could benefit from a single sentence to at the end summarising the competing effects and which one takes over from which one leading to each pattern observed.

A2.18. Thank you for the opportunity to clarify the findings. At the end of each section, we added the following three summarizing statements:

Lines 397-400: “To summarize, the N_F cell’s response to the weak ac field is through the oscillations of **P** with the frequency of the applied field. There is no voltage threshold for the occurrence of oscillations. As the voltage increases past a threshold U_{ST} , a new stripe structure emerges in which the oscillations are centered at the stationary splay and bend deformations of **P**.”

Lines 485-490: “To summarize, the splay-twist stripes are caused by the balance of dielectric and elastic torques. The field-induced splay in the vertical cross-section of the cell is reduced by the in-plane splay of an opposite polarity. This electrostatic splay cancellation produces patterns with a period much larger than the thickness of the cell. As the voltage increases further, the one-dimensional deformations in the plane of the cell are replaced by two-dimensional deformations with a square lattice of -1 and +1 defects; the latter are of splay geometry.”

Lines 519-523: “The square lattice thus exhibits simultaneously numerous mechanisms of coupling of the electric field to the liquid crystal structure: oscillations of polarization, dielectric-elastic balance that produces stationary splay and bend, accumulation of stationary potential differences at the electrodes of the cell as a result of deformations produced by a high-frequency ac voltage with zero dc component, and, finally, occurrence of hydrodynamic flows.”

R2.19. Figures 1 and 2 feel like they could be arranged more clearly to allow for an easier reading experience for the reader. It very much feels like you have to bounce all over the figure and leading to the feeling that they are not called in the correct order in the text (i.e. 1b before 1a). I don't believe they are called in an incorrect order in the text but the sensation of having to continuously look back at previous figures makes the paper more complicated to follow.

A2.19. Thank you, we rearranged the figures by moving some figures previously presented as Supplement and adding new experimental results.

Reviewer #4

R3.1. Frederiks transition in a ferroelectric nematic liquid crystalline phase (NF) is reported under the influence of an ac electric field. In ferroelectric nematics, no Frederiks transition is observed if a dc field is applied to planar cells filled with liquid crystal in the NF phase, because polarization in the NF phase is so high that the external dc field can always be compensated by a rotation of polarization (which means rotation of the average direction of the nematic director, because polarization P is along the local average direction of the long molecular axes). On the other hand, when an ac field is applied and the amplitude is increased, authors observe three behavioural regimes: a homogeneous rotation at low amplitudes, periodic stripes of splay and twist deformation of P , and, at high enough amplitudes, periodic square lattice with splay and bend of P . The paper offers a thorough identification of the NF structure in all three regimes by using several experimental techniques.

A3.1. We thank the Reviewer for stressing a thorough identification of the structure. We added new data on the dynamics of these structures in the revised manuscript.

R3.2. However, it does not address the origin/driving force for the formation of such structures. We would like to point out, that similar texture changes were reported also in conventional nematics, where a variety of pattern morphologies may arise due to the flexoelectricity and electroconvection at high ac field frequency (such affects were extensively studied by the group of A. Buka, see for example *Liquid Crystals Reviews*, 4:2, 101-134 (2016), by N. Éber et al, and also the paper by Sasaki et al in *Nat. Commun.* in 2016; DOI: 10.1038/ncomms13238). How and why were these mechanisms excluded? For conventional nematics, it was demonstrated that the AC voltage induces periodic patterns for example due to the ions accumulated at an electrically insulating polymer interface. Such one-dimensional and two-dimensional arrays of defects are particularly common for nematic phase with an ionic impurity. In the NF phase, as pointed out also by authors of the present manuscript, the ionic impurities are quite high, so self-organization of arrays of defects is quite probable. Based on the above, we cannot recommend the acceptance of the paper in this form. However, if the authors can include in the manuscript a plausible explanation on the difference between the effect reported in conventional nematics and the effects leading to the formation of complex patterns that they observed in ferroelectric nematics, the paper can be considered for *Nature Communications*.

A3.2. We agree with the Reviewer that the mechanisms involved are multifaceted. We discuss these mechanisms in the revised manuscript. We cite the papers by Eber et al ⁸ and by Sasaki et al. ⁹. Although the patterns are similar, there are crucial differences. The stripe and square patterns have been reported by Sasaki et al in a dielectrically negative material CCN-37 with a high concentration of added ions. Yet, the frequencies at which these patterns occur are limited from above, as Sasaki et al. convincingly show, and this limit is less than 200-300 Hz. In our case, the patterns are observed in a dielectrically positive material at frequencies that are thousand times higher than this limit. The period of patterns also differ by one order of magnitude. More importantly, there are principal differences in the mechanisms.

First, we demonstrate that the response in the homogeneous and deformed states is accompanied by the oscillating polarization, with the frequency of the applied field, which is a unique feature of the NF material never before observed in the conventional N. This feature is not affected by the ions, as we described in the Discussion section, lines 392-396:

“Free ions do not affect the dynamic patterns at the high frequencies of the field explored in this work. In the denominator of Eq. (6), the first term $P^2 \cos^2 \bar{\psi} \approx P^2$ is much larger than the imaginary part of the second term, which writes $\gamma \sigma_{\perp}$; with the rotational viscosity $\gamma = 5 \text{ Pa} \cdot \text{s}^2$ and $\sigma_{\perp} \sim 10^{-7} \text{ S/m}$, one estimates $\frac{P^2}{\gamma \sigma_{\perp}} \sim 10^4$, which supports the statement that the free ions do not play a role in the oscillatory regime.”

Second, the estimates of flexoelectric effect show that it is much smaller than the effect of bound charges, line 474-477:

“For the typical $e_1 = 10^{-11} \text{ C/m}$ (and even for a few orders of magnitude larger e_1 , Ref. ⁷), $P = 6 \times 10^{-2} \frac{\text{C}}{\text{m}^2}$ and experimentally observed $L \approx 3 \times 10^{-4} \text{ m}$, one concludes that the flexoelectric charge in the explored patterns is smaller than the polarization charge, $[\rho_f / \rho_b] =$

$$\left| \frac{\pi}{2P} e_1 \left(\frac{\bar{\psi}}{\lambda} - \frac{2\bar{\varphi}_m}{L} \right) \right| \leq \left| \frac{\pi \bar{\varphi}_m}{LP} e_1 \right| \sim 10^{-5}.”$$

Third, we demonstrate that the application of the ac field with zero bias creates a stationary potential difference at the electrodes, which we connect to the bound charges and which we consider as the reason for electrohydrodynamics. This is different from the distribution of free charges in the case of Sasaki et al. patterns. We do not exclude the electrohydrodynamics, as the dynamic experiments demonstrate electrohydrodynamic flows of the material in the square lattices, illustrated in the added Fig.6e.f. To stress the differences, as required by the Reviewer, we added a paragraph before the Discussion section, lines 331-342:

“The large-period (as compared to the slab thickness) stripes and square pattern of splay-bend resemble the stripes and square lattices in field-free hybrid aligned N films ¹⁰ and in the geometry of the bend Fréedericksz transition in a material of a negative $\Delta\epsilon$ intentionally doped

with a large amount of ions⁹. In the latter case, the patterns are observed below some limiting frequency of the applied field, which is a few hundreds of Hz, a thousand times lower than in our case. The stripes appear at higher voltages than the square lattices; the behavior is opposite in our case. The periodicities are about one order of magnitude lower than in our case. We attribute the differences to the different nature of operational charges in the two works. Sasaki et al.⁹ explain the observed patterns by the nonuniform distribution of freely moving ions that accumulate at the insulating polymer alignment layers. In our case, the main driving mechanism are bound charges which in the case of stripes tend to geometrically self-compensate and in the case of the square lattices, produce a stationary potential difference between the electrodes.

”

R3.3. We also have some comments regarding the text and figures:

Line 63: The threshold voltage depends on the cell thickness, so we suggest that the cell thickness is given in the text as well (not only in the figure caption).

A3.3. Thank you, we added the thickness values whenever possible.

R3.4. Line 196: titled -> tilted

A3.4. Thank you, we corrected this misprints.

R3.5. Figures:

Figures are very well prepared and rich in information. However, we suggest that the authors consider the following comments:

- 1) The size of the text on the axes should be increased in all figures both in the main paper and the extended data.
- 2) For several figures, the symbols used in the figures are not introduced in the figure caption.

R3.5. Thank you, we revised the figures by increasing the fonts and explained the symbol.

R3.6. Additional comments to specific figures:

- 1) Figure 1: different structures are observed in the same cell just by increasing the voltage amplitude. What is the reason for the figure parts being given for three different cells of different thickness? Also, explain in the caption what are P and R in the figure.
- 2) Extended Data Figure 5: amplitude of the field -> amplitude of the voltage (U)

R3.6. 1) Figure 1: The main reason was to demonstrate the validity of results at different cell thicknesses. We streamlined the presentation and used the same cells for multiple panels in the figures. We explained P and R in the captions.

2). Thank you, we corrected the misprint.

References used in the Reply:

- 1 Clark, N. A., Chen, X., MacLennan, J. E. & Glaser, M. A. Dielectric spectroscopy of ferroelectric nematic liquid crystals: Measuring the capacitance of insulating interfacial layers. *Physical Review Research* **6**, 013195 (2024).
<https://doi.org/10.1103/PhysRevResearch.6.013195>
- 2 Adaka, A. *et al.* Dielectric Properties of a Ferroelectric Nematic Material: Quantitative Test of the Polarization-Capacitance Goldstone Mode. *Phys Rev Lett* **133**, 038101 (2024). <https://doi.org/10.1103/PhysRevLett.133.038101>
- 3 Adaka, A. *et al.* Dielectric Properties of a Ferroelectric Nematic Material: Quantitative Test of the Polarization-Capacitance Goldstone Mode. *Phys Rev Lett* **133**, 038101 (2024). <https://doi.org/10.1103/PhysRevLett.133.038101>
- 4 Erkoreka, A. & Martinez-Perdiguero, J. Constraining the value of the dielectric constant of the ferroelectric nematic phase. *Phys Rev E* **110**, L022701 (2024).
<https://doi.org/10.1103/PhysRevE.110.L022701>
- 5 Chen, X. *et al.* First -principles experimental demonstration of ferroelectricity in a thermotropic nematic liquid crystal: Polar domains and striking electro-optics. *Proceedings of the National Academy of Sciences of the United States of America* **117**, 14021-14031 (2020). <https://doi.org/10.1073/pnas.2002290117>
- 6 Basnet, B. *et al.* Soliton walls paired by polar surface interactions in a ferroelectric nematic liquid crystal. *Nature Communications* **13**, 3932 (2022).
<https://doi.org/10.1038/s41467-022-31593-w>
- 7 Barthakur, A., Karcz, J., Kula, P. & Dhara, S. Critical splay fluctuations and colossal flexoelectric effect above the nonpolar to polar nematic phase transition. *Phys Rev Mater* **7**, 035603 (2023). <https://doi.org/10.1103/PhysRevMaterials.7.035603>
- 8 Éber, N., Salamon, P. & Buka, Á. Electrically induced patterns in nematics and how to avoid them. *Liquid Crystals Reviews* **4**, 102-135 (2016).
<https://doi.org/10.1080/21680396.2016.1244020>
- 9 Sasaki, Y. *et al.* Large-scale self-organization of reconfigurable topological defect networks in nematic liquid crystals. *Nature Communications* **7**, 13238 (2016).
<https://doi.org/10.1038/ncomms13238>
- 10 Lavrentovich, O. D. & Pergamenschchik, V. M. Patterns in Thin Liquid-Crystal Films and the Divergence (Surfacelike) Elasticity. *Int J Mod Phys B* **9**, 2389-2437 (1995).
<https://doi.org/10.1142/S0217979295000926>

Respectfully yours,

Oleg D. Lavrentovich,

Reply to Reviewers

We thank all the reviewers for helpful suggestions and questions which helped us to improve the presentation. We list our replies below for the second round of reviews.

Reviewer #1 (Remarks to the Author):

R1.1. The revised paper is sound and suitable for publication in its current format, with a few final comments from us below.

Line 121-124. There are lots of inline equations in this section which I found harder to follow with them being inline. Possibly consider inputting them as display equations instead. Though this is just a suggestion for readability.

A1.1. Thank you, we presented the long formula as a display equation

R1.2. Line 125 could you state the “such a way” the compensators were used to aid in repeatability. Presumably both Γ and Γ_0 are related to the sum of both the retardance of the cell and the compensator here? Or how then does $2\pi/\lambda * \Gamma_0 = \pi/2$?

A1.2. Yes, the statement is correct, we clarified the issue by adding a sentence on line 128:

“Both $\Gamma(U)$ and Γ_0 represent a sum of the retardances of a cell and an optical compensator.”

R1.3. Fig 2b,c clearly show some phase shift between the applied voltage and optical response. Fig4b,c shows maybe a smaller phase shift or maybe none at all? It is hard to tell from the figure and maybe a vertical line as in fig2 could be used here. Would the considerations to the phase difference from fig2 apply equally to fig 4?

A1.3. We added a clarification, lines 309-312: “Note that there is some phase shift between the applied voltage and the optical response in Fig.4b, similarly to the case presented in Fig.2, for a qualitatively similar reason. However, the stripe pattern is strongly inhomogeneous, and the analysis of the shift is less transparent and will require additional studies.”

R1.4. Fig.6c. remark the “c” so it is not obscured. I also feel it would be useful to plot what the voltage profile applied by the generator is to show how distinct this change is.

A1.4. Thank you, we made “c” visible. We also added a clarification in the text, lines 341-342 ” despite the fact that the generator produces a standard sinusoidal waveform with zero dc bias.”

Reviewer #2 (Remarks to the Author):

R2.1. I co-reviewed this manuscript with one of the reviewers who provided the listed reports. This is part of the Nature Communications initiative to facilitate training in peer review and to provide appropriate recognition for Early Career Researchers who co-review manuscripts.

A2.1. We thank the Reviewer and co-Reviewer for the useful comments.

Reviewer #4 (Remarks to the Author):

R4.1. The authors made extensive improvements of the manuscript regarding the clarity and they also included additional measurements to support their findings. All the points that were raised in the first report were considered. We thus recommend the paper to be accepted for publication.

A4.1. We thank the Reviewer(s) for the useful comments.